# Study on the Performance of Recycled Cement-Stabilized Macadam Mixture Improved Using Alkali-Activated Lithium Slag–Fly Ash Composite

**Weijun Yang** [1,2]**, Zhenzhou Jin** [1]**, Jianyu Yang** [1,]*****, Jiangang He** [2]**, Xuemei Huang** [3]**, Xin Ye** [1]**, Guangyao Li** [3] **and Chao Wang** [3]

[1] School of Civil Engineering, Changsha University of Science and Technology, Changsha 410114, China; yyyaozhijian@163.com (W.Y.); 17816752813@163.com (Z.J.); yxx@stu.csust.edu.cn (X.Y.)

[2] Hunan Renjian Baogu High-Tech Development Co., Changsha 410114, China; kaigao0920@163.com

[3] School of Transportation Engineering, Changsha University of Science and Technology, Changsha 410114, China; huangxuemei_03@163.com (X.H.); ligy1217@163.com (G.L.); wcxt@stu.csust.edu.cn (C.W.)

***** Correspondence: jianyuy@csust.edu.cn; Tel.: +86-13873191702

**Abstract:** The huge demand for sand and gravel resources in road engineering construction leads to excessive consumption of resources and environmental damage. Recycling waste concrete and industrial solid waste as a road material is a promising alternative. In order to explore the application of these solid wastes in the road base, this paper studies the effect of adding lithium slag activated by an alkaline activator, fly ash (FA) and a combination of the two on the compressive strength, splitting strength and shrinkage performance of recycled cement-stabilized macadam mixture (RCSM). The optimum content of recycled aggregate (RA), alkali-activated lithium slag (AALS) and FA in composite-improved RCSM was optimized using a response surface method (Box–Behnken), and the microscopic characteristics of the mixture were analyzed using X-ray diffraction (XRD) and scanning electron microscopy (SEM). The results show that the optimum dosage of AALS, FA and RA determined by the response surface method is 15%, 10% and 40%, respectively. Compared with the cement-stabilized macadam mixture (CSM) with 40% RA, the 28 d compressive strength and 28 d splitting strength of the composite-improved RCSM are increased by 26.8% and 22.9%, respectively, and the dry shrinkage coefficient and average temperature shrinkage coefficient are decreased by 25.8% and 14.8%, respectively. Microscopic tests show that AALS and FA participate in the hydration reaction, generate more hydrated silicate (C-S-H) and ettringite (AFt), refine pores, effectively improve the performance of the internal interface transition zone of the mixture, make the microstructure of the mixture denser, and improve the strength and shrinkage performance of RCSM. This study provides technical support for the reuse of resources and the sustainable development of road construction.

**Keywords:** cement-stabilized macadam mixture; recycled aggregates; alkali-activated lithium slag; fly ash; response surface methodology; microscopic properties





## 1. Introduction

During the renovation and reconstruction of damaged cement concrete pavements, a large amount of discarded concrete slabs are generated. If these slabs are discarded directly, they will not only occupy valuable land resources, resulting in a severe waste of resources, but also bring unnecessary economic losses due to the failure to fully utilize these discarded slabs. More critically, the chemical substances contained in the discarded concrete may gradually seep into the soil and groundwater, posing a long-term and irreversible pollution risk to the ecological environment [1–3]. Therefore, how to achieve efficient recycling and utilization of these discarded concretes has become a current research focus.

In recent years, researchers have been dedicated to crushing and processing discarded concrete into recycled aggregate (RA) and exploring its application in cement-stabilized macadam base. This practice not only helps to improve the mechanical properties of the base but also promotes the recycling and utilization of waste mineral resources, achieving the dual goals of resource recycling and environmental protection. However, due to the crushing process of RA, they are subjected to rolling and crushing, resulting in many weak points under stress, leading to poor homogeneity and an uneven distribution of weak points. These problems will lead to a decrease in the strength of recycled cement-stabilized macadam mixture (RCSM), so it is necessary to strictly control the dosage, particle size and other parameters of RA to ensure the strength requirements of the mixture. How to optimize its application effect is still the focus of current research [4–7].

Fly ash (FA), a solid waste generated from coal combustion in thermal power plants, mainly contains silicon dioxide and alumina as its chemical components. For a long time, this solid waste has not been fully utilized. However, studies have found that adding an appropriate amount of FA to cement-stabilized macadam mixtures (CSMs) can effectively improve the mechanical properties of the mixtures [8,9]. China is rich in lithium ore resources, with an estimated reserve of approximately 1.1 million tons of extractable lithium ore. Lithium slag, as a by-product of lithium carbonate extraction from spodumene, exhibits good pozzolanic activity and micro-aggregate filling performance after activation by alkaline activators, making it a potential building material [10]. Mixing alkali-activated lithium slag (AALS) and FA into recycled cement-stabilized macadam base can not only achieve effective waste treatment but can also significantly enhance the mechanical properties and durability of the recycled base by leveraging their excellent properties.

Many researchers have carried out research on the strength and durability of recycled cement-stabilized macadam base. Disfani et al. [11] evaluated the performance of RCSM. The results show that when the cement content is 5% and the RA content is less than 50%, the mechanical properties and fatigue durability of the recycled mixture meet the requirements of the specification. Lan et al. [12,13] used RA as cement-stabilized macadam base material. Initially, due to the hydration reaction of unhydrated cement mortar on the surface of RA, the strength and resilient modulus of the base were improved. However, with the increase in RA content, the decrease in the strength of RA itself is greater than the strength provided by cement hydration reaction, and the strength and resilient modulus of the base decrease. Yan et al. [14,15] found that the RA has large porosity and high water absorption. With the increase in RA content, the dry shrinkage coefficient of the mixture continues to increase. You et al. [16,17] used physical and chemical methods to treat the surface of RA and used the treated RA to prepare RCSM. The strength, stiffness and shrinkage performance of recycled mixture were improved. Deng et al. [18–20] found that the incorporation of FA into CSM can effectively improve the internal structure of the mixture and improve the mechanical properties of the mixture by using its micro aggregate filling effect. Zhang et al. [21] added alkali-activated steel slag to RCSM and tested its mechanical properties and durability. The results showed that the addition of alkali-activated steel slag promoted the hydration reaction and generated additional cementitious products, which significantly improved the compressive strength, frost resistance and shrinkage performance of the mixture. He et al. [22] found that lithium slag will undergo secondary hydration reaction with cement hydration products to form hydrated silicate and ettringite. These hydration products can effectively fill the pores inside the concrete, thereby improving the later strength of the concrete. Through a single-factor test, it is concluded that the optimum content of lithium slag in concrete is 20%. Qin et al. [23] found that the cube compressive strength and elastic modulus increased by 9.9% and 9.94% when 20% lithium slag and 30% recycled coarse aggregate were added to concrete through laboratory tests. The reason is that the incorporation of lithium slag promotes the development of concrete cementitious structure, improves its microstructure and refines the pores. In summary, scholars have actively explored the performance of RCSM. Although some studies have focused on the improvement of the performance of RCSM using admixtures, there is still insufficient

research on the utilization of industrial solid waste, especially lithium slag and FA. In view of this situation, this paper is devoted to making up for the shortcomings of existing research, deeply exploring the effects of two kinds of industrial solid wastes, lithium slag and FA, on the mechanical properties and durability of RCSM, and analyzing its mechanism of action, to promote the green and sustainable process of road construction.

This article aims to improve the mechanical properties of recycled cement-stabilized macadam base, minimize the drying shrinkage and temperature shrinkage of the base, make RA and industrial solid waste more widely used in road base, make full use of the value of waste, and reduce the environmental pollution caused by construction waste. The study investigated the effects of three measures, namely adding AALS, adding FA, and adding both AALS and FA, on the compressive strength, splitting strength, and shrinkage performance of RCSM. The response surface method (Box–Behnken) was used to optimize the optimal dosages of RA, AALS, and FA in the composite-improved RCSM, and the microstructural characteristics of the composite-improved RCSM were analyzed. This study not only provides important theoretical and technical support for the promotion and application of RCSM but also contributes new solutions to the effective utilization of waste concrete and the reduction in environmental pollution.

## 2. Materials and Methods

### 2.1. Materials

The cement used was P.C42.5 composite Portland cement. The natural aggregate was limestone, and all the indexes met the requirements of the specification [24]. The RA was made from the crushing of the waste concrete pavement in Xiangtan, Hunan Province. In addition to the water absorption rate, the physical and mechanical properties of the RA met the requirements of the specification [24] for natural aggregate. Lithium slag was selected from the industrial waste after the production of lithium salt by Jiangxi Nan's Lithium Materials Co. (Yichun, China). The alkali activator was water glass with a modulus of 3.3, and solid NaOH with a purity of 98% was used to adjust the modulus of the water glass, with a modulus of 1.5. Water glass constituted 6% of the lithium slag. Grade II FA was provided by Henan Hengyuan New Materials Co. (Xinyang, China). The relevant technical indicators are shown in Tables 1–6.

**Table 1.** Technical indexes of cement.

| Test Items | Technical Indicators | Specification Requirements |
|---|---|---|
| Specific surface area/$m^2 \cdot kg^{-1}$ | 362 | $\geq$300 |
| Initial setting time/min | 82 | $\geq$45 |
| Final setting time/min | 183 | $\leq$600 |
| Compressive strength (28 d)/MPa | 49.3 | $\geq$42.5 |
| Flexural strength (28 d)/MPa | 8.3 | $\geq$6.5 |

**Table 2.** Chemical composition of cement.

| $SiO_2$ | $Fe_2O_3$ | $Al_2O_3$ | $SO_3$ | MgO | CaO |
|---|---|---|---|---|---|
| 22.1% | 3.7% | 6.6% | 3.6% | 1.1% | 56.9% |

**Table 3.** Road performance index of natural coarse aggregate and recycled coarse aggregate.

| Test Items | Natural Coarse Aggregate | Recycled Coarse Aggregate | Test Method | Specification Requirements |
|---|---|---|---|---|
| Apparent relative density/($g \cdot cm^{-3}$) | 2.689 | 2.681 | T 0308—2005 | $\geq$2.50 |
| Water absorption/% | 0.93 | 3.75 | T 0308—2005 | $\leq$3.0 |
| Silt content/% | 0.8 | 1.3 | T 0310—2005 | $\leq$1.5 |
| Needle-like particle content/% | 6.8 | 14.2 | T 0312—2005 | $\leq$18 |
| Ruggedness/% | 5.1 | 9.6 | T 0314—2024 | $\leq$12 |
| Crushing value/% | 19.5 | 23.2 | T 0316—2024 | $\leq$26 |
| Los Angeles abrasion loss/% | 14.8 | 26.5 | T 0317—2005 | $\leq$30 |

**Table 4.** Road performance index of natural fine aggregate and recycled fine aggregate.

| Test Items | Natural Fine Aggregate | Recycled Fine Aggregate | Test Method | Specification Requirements |
|---|---|---|---|---|
| Apparent relative density/(g·cm$^{-3}$) | 2.793 | 2.569 | T 0330—2005 | ≥2.50 |
| Water absorption/% | 13.3 | 24.2 | T 0330—2005 | — |

**Table 5.** Chemical composition of lithium slag.

| SiO$_2$ | Fe$_2$O$_3$ | Al$_2$O$_3$ | SO$_3$ | MgO | CaO |
|---|---|---|---|---|---|
| 39.0% | 2.6% | 18.0% | 9.2% | 1.0% | 8.4% |

**Table 6.** Chemical composition of FA.

| SiO$_2$ | Al$_2$O$_3$ | CaO | Fe$_2$O$_3$ | MgO | SO$_3$ |
|---|---|---|---|---|---|
| 50.5% | 25.6% | 9.8% | 6.9% | 1.1% | 1.1% |

### 2.2. Mixture Ratio

According to Reference [25], to improve the mechanical properties of RCSM, the cement mass fraction used in the laboratory test was 5%. To meet the technical requirements of China's specification [26] for the first-class highway, the C-B-3 gradation median of the skeleton dense structure was selected for the gradation, and the range of the reference gradation is shown in Table 7.

**Table 7.** RCSM gradation.

| Gradation | Mass Percentage Passing the Following Sieve/% | | | | | | |
|---|---|---|---|---|---|---|---|
| | 31.5 mm | 19 mm | 9.5 mm | 4.75 mm | 2.36 mm | 0.6 mm | 0.075 mm |
| Upper limit of grading | 100 | 86.0 | 58.0 | 32.0 | 28.0 | 15.0 | 3.0 |
| Lower limit of grading | 93.0 | 68.0 | 38.0 | 22.0 | 16.0 | 8.0 | 0 |
| Design gradation | 100 | 77.0 | 48.0 | 27.0 | 22.0 | 11.5 | 1.5 |

The compaction test was carried out according to the Chinese standard (JTG E51-2009) [27]. The optimum moisture content (OMC) and maximum dry density (MDD) are shown in Table 8.

**Table 8.** OMC and MDD.

| AALS-Improved Mixture | | | | FA-Improved Mixture | | | | Composite-Improved Mixture * | | | |
|---|---|---|---|---|---|---|---|---|---|---|---|
| AALS/% | RA/% | OMC/% | MDD/g·cm$^{-3}$ | FA/% | RA/% | OMC/% | MDD/g·cm$^{-3}$ | AALS/% | FA/% | OMC/% | MDD/g·cm$^{-3}$ |
| 10 | 25 | 5.8 | 2.305 | 5 | 25 | 5.8 | 2.301 | 10 | 5 | 6.6 | 2.283 |
| 10 | 40 | 6.4 | 2.287 | 5 | 40 | 6.5 | 2.275 | 15 | 5 | 6.7 | 2.278 |
| 10 | 55 | 6.8 | 2.271 | 5 | 55 | 7.1 | 2.238 | 20 | 5 | 6.8 | 2.273 |
| 15 | 25 | 5.9 | 2.298 | 10 | 25 | 6.1 | 2.291 | 10 | 10 | 6.8 | 2.271 |
| 15 | 40 | 6.4 | 2.285 | 10 | 40 | 6.7 | 2.275 | 15 | 10 | 6.8 | 2.265 |
| 15 | 55 | 6.8 | 2.268 | 10 | 55 | 7.2 | 2.235 | 20 | 10 | 6.9 | 2.255 |
| 20 | 25 | 6.0 | 2.293 | 15 | 25 | 6.2 | 2.289 | 10 | 15 | 6.9 | 2.258 |
| 20 | 40 | 6.5 | 2.279 | 15 | 40 | 6.8 | 2.273 | 15 | 15 | 7.0 | 2.243 |
| 20 | 55 | 6.9 | 2.247 | 15 | 55 | 7.4 | 2.226 | 20 | 15 | 7.0 | 2.241 |

* The RA content of composite-improved mixture is 40%.

### 2.3. Strength Test

This study strictly follows the relevant requirements of the Chinese code (JTG E51-2009) [27]. Cylindrical specimens with diameters of 150 mm and heights of 150 mm were

used for the unconfined compressive strength test, splitting strength test and compressive modulus of elasticity test. The flexural strength test was carried out using the middle beam specimens with a length of 100 mm × width of 100 mm × height of 400 mm. The specimens were maintained in a standard curing room (temperature 20 ± 2 °C, relative humidity ≥ 95%) for 27 days, and the specimens were immersed in water on the last day. After curing, the mechanical properties were tested according to the specified loading rate. Five parallel tests were carried out for each mix ratio. The unconfined compressive strength test was carried out by hydraulic press, and other mechanical tests were carried out using an MTS universal testing machine. The loading rate of unconfined compressive strength test, splitting strength test and compressive modulus of elasticity test was 1 mm/min, and the loading rate of flexural strength test was 50 mm/min. The test process is shown in Figure 1.

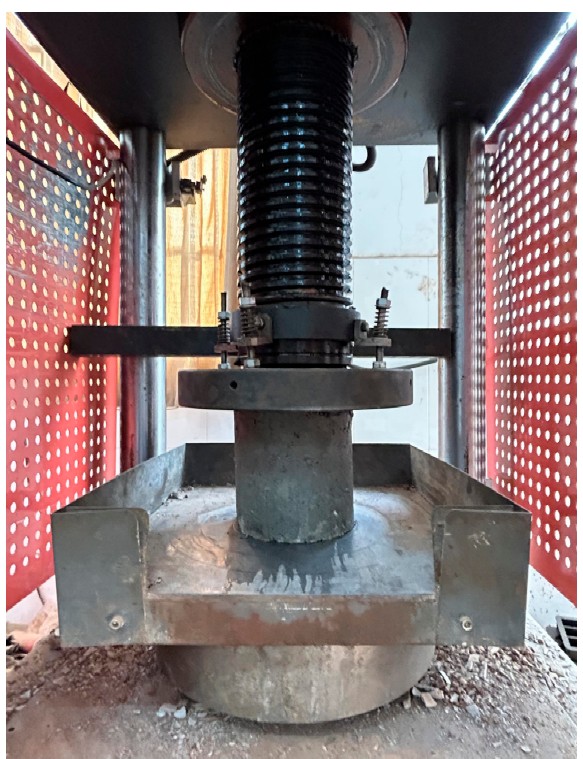

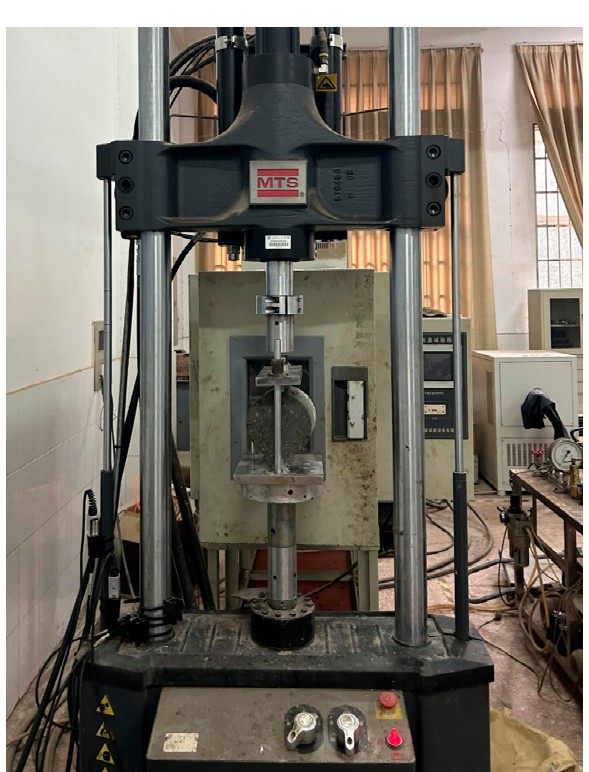

（a）Unconfined compressive strength test　　　　　　　　　　　　（b）Splitting strength test

**Figure 1.** Strength test.

### 2.4. Dry Shrinkage/Temperature Shrinkage Test

According to the relevant requirements of the Chinese code (JTG E51-2009) [27], the dry shrinkage and temperature shrinkage tests were carried out on the middle beam specimens with a length of 100 mm × width of 100 mm × height of 400 mm. The middle beam specimens were cured in the standard curing room for 6 days, and the specimens were soaked in water on the last day. After curing, the dry shrinkage and temperature shrinkage tests were carried out, and three parallel tests were carried out for each mix ratio. The drying shrinkage test was carried out at a temperature of (20 ± 1) °C and a relative humidity of (60 ± 5)%. The temperature ranges of the temperature shrinkage test were 40~30 °C, 30~20 °C, 20~10 °C, 10~0 °C, and 0~10 °C. The test was carried out step by step from a high to low temperature, each stage was lasted 3h, and the cooling rate was 0.5 °C/min. The experiment is shown in Figure 2.

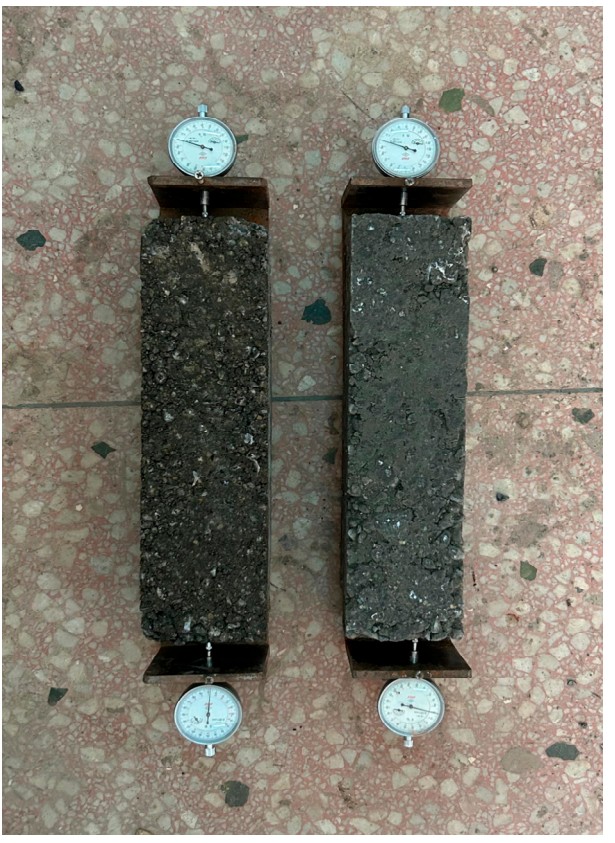

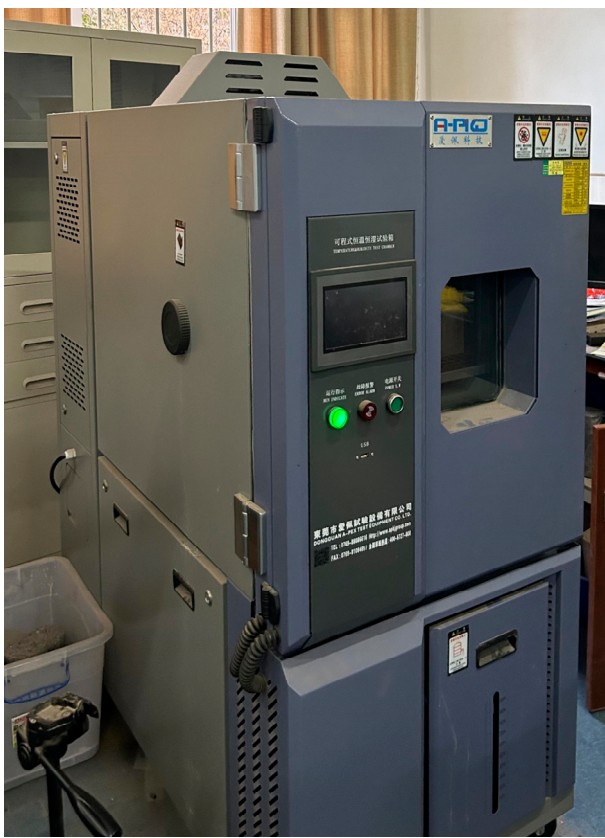

（a）Dry Shrinkage Test　　　　　　　　（b）Temperature Shrinkage Test

**Figure 2.** Dry shrinkage/temperature shrinkage test.

*2.5. Microscopic Tests*

The XRD diffraction test of the sample was carried out using a Bruker D8 advance powder X-ray Cu Ka radiation calibrator (Bruker, Billerica, MA, USA). The relevant parameters of the test were a wavelength of 0.15406 nm, voltage of 40 KV, and current of 40 mA. SEM was carried out using a German ZEISS Sigma 300 (Carl Zeiss AG, Oberkochen, Germany). Before the test, the sample was plated with gold-palladium alloy to increase its conductivity.

## 3. Results and Discussion

*3.1. Performance Test*

3.1.1. Strength Test

An RCSM was prepared by incorporating RA, and the effects of adding AALS and FA on the compressive strength and splitting strength of the RCSM were analyzed. On the basis of controlling the optimal content of RA, the effects of AALS and FA on the performance improvement of the RCSM were studied. A control group of pure 40% RA mixture was set up. The test results are shown in Figure 3.

From Figure 3a,b,d,e, it can be seen that (1) when the amount of AALS and FA remains constant, the strength of the RCSM increases first and then decreases with the increase in the RA content. When the content of RA reaches 40%, the strength reaches its maximum value. This is because, when the content of RA is less than 40%, the unhydrated old cement mortar on the surface of the RA reacts with water, strengthening the bonding force inside the mixture, thus increasing the strength of the mixture. However, as the content of RA continues to increase, the decline in the strength of the RA itself exceeds the strength generated by cement hydration, so the strength of the mixture gradually decreases. (2) When the content of RA remains constant, with the increase in AALS content,

the unconfined compressive strength and splitting strength of the RCSM first increase and then decrease. When the content of AALS is approximately 15%, the maximum unconfined compressive strength and splitting strength reach 6.24 MPa and 0.52 MPa, respectively. This is because AALS and cement hydration products undergo secondary hydration reactions, generating calcium silicate hydrate and ettringite, which can refine pores, greatly optimize the performance of interface transition zones, and enhance the overall density, thus improving the unconfined compressive strength and splitting strength of the RCSM. In addition, more irregular petal-like hydrated calcium silicate and prismatic ettringite were observed in the microscopic test, which can effectively verify the above reaction and performance improvement process. (3) Adding FA significantly improves the compressive strength and splitting strength of the mixture. The optimal content is 10%, and the maximum 28 d unconfined compressive strength and 28 d splitting strength reach 6.6 MPa and 0.56 MPa, respectively. This is mainly due to two functions of FA in the RCSM: one is to fill gaps; the other is to exert its volcanic ash activity. FA reacts with cement to form cementitious products, increasing the bonding force between particles and thus improving the strength of the RCSM.

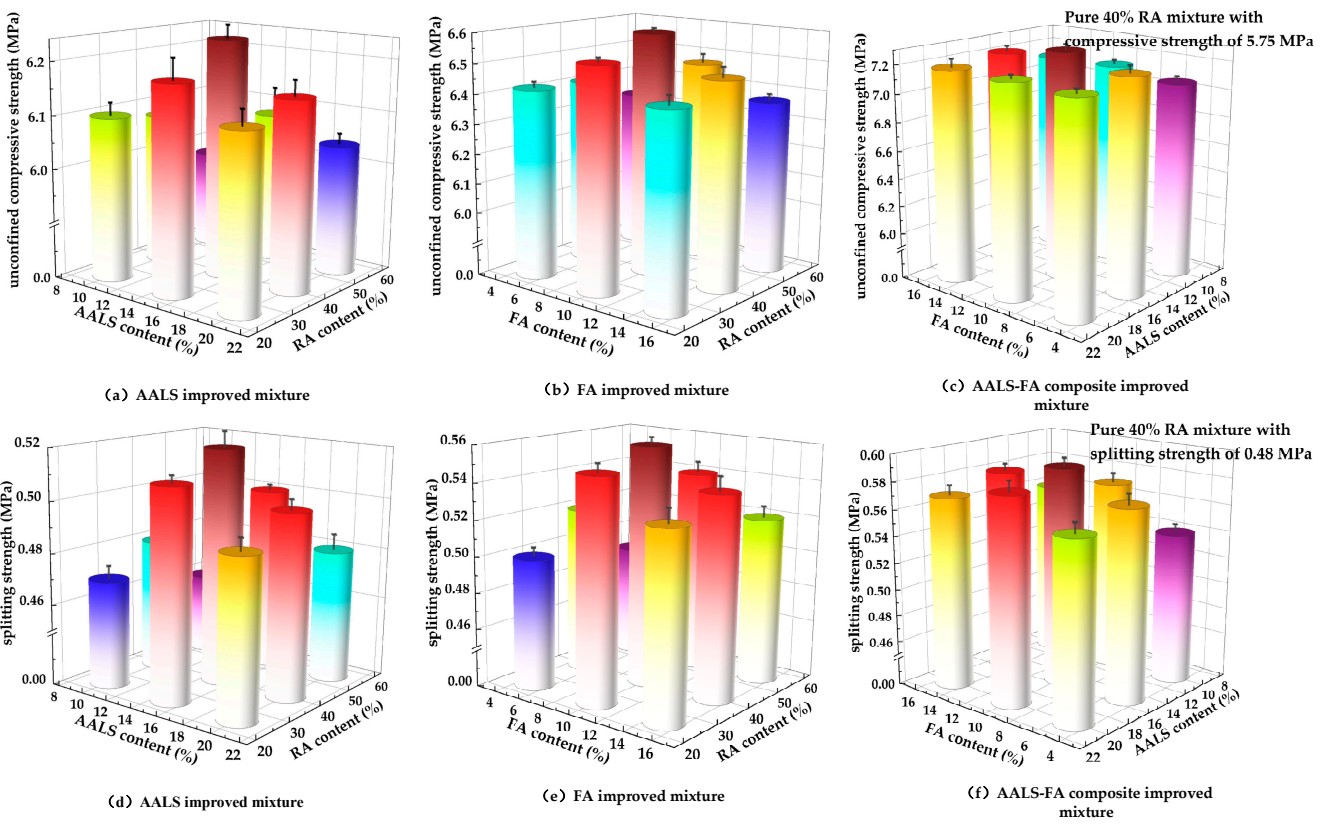

**Figure 3.** Unconfined compressive strength test results and splitting strength test results.

From Figure 3c,f, it can be seen that by mixing 40% RA to RCSM and adding AALS and FA to the mixture, the unconfined compressive strength and splitting strength of the RCSM can be significantly enhanced. When the contents of AALS and FA are 15% and 10%, respectively, the 28 d compressive strength and 28 d splitting strength reach the maximum values of 7.29 MPa and 0.59 MPa, respectively, which are 26.8% and 22.9% higher than those of the pure mixed mixture with 40% RA.

### 3.1.2. Drying/Temperature Shrinkage Test

The preparation of RCSM by incorporating RA, and the effects of adding AALS, FA, and an AALS–FA compound on the improvement of the shrinkage performance of the

RCSM were analyzed. A control group consisting of a pure 40% RA mixture was set up. The test results are shown in Figure 4.

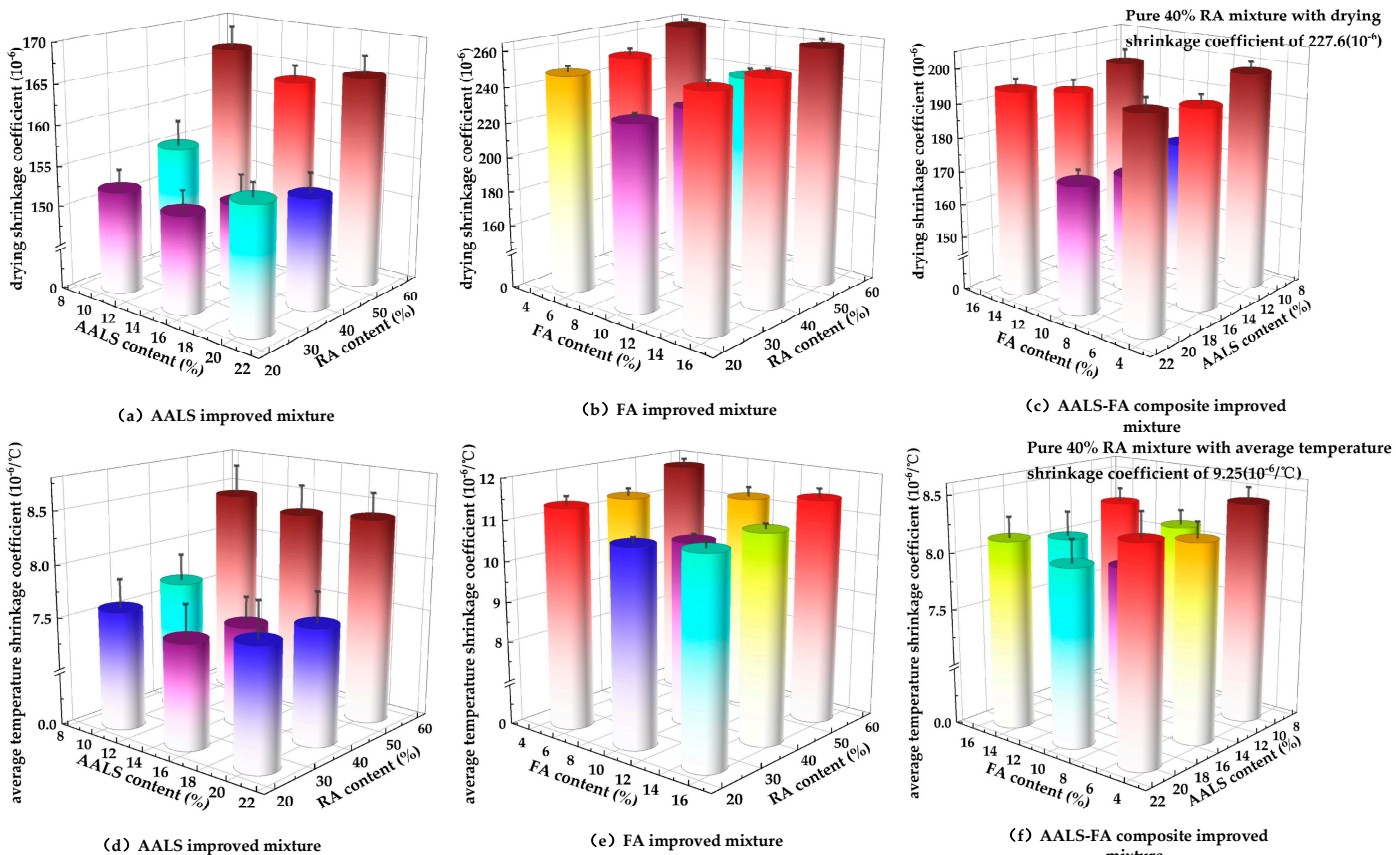

**Figure 4.** Dry shrinkage/temperature shrinkage test results.

From Figure 4a,b,d,e, it can be seen that (1) when the contents of AALS and FA remain constant, the shrinkage coefficient and average temperature shrinkage coefficient of the RCSM increase slowly with the increase in the RA content. However, when the content exceeds 40%, the shrinkage coefficient and average temperature shrinkage coefficient increase rapidly. After the test, it was observed that with the increase in RA content, the number and width of cracks on the surface of the sample gradually increased. This is because the surface of the RA has many pores, with a high water absorption rate and rapid water loss, leading to an increase in the shrinkage coefficient and temperature shrinkage coefficient of the mixture. The effect is more significant when the content of RA exceeds 40%. (2) When the content of RA remains constant, with the increase in AALS content, the shrinkage coefficient and average temperature shrinkage coefficient of the RCSM first decrease and then increase. When 15% AALS is added to the pure mixed mixture with 40% RA, the shrinkage coefficient and average temperature shrinkage coefficient of the mixture decrease by 33.9% and 19.7%, respectively. This is because lithium slag contains silicon dioxide, alumina, and a small amount of sulfur trioxide. When it reacts with calcium hydroxide, a product of cement hydration, ettringite is formed, leading to a micro-expansion effect. This micro-expansion increases the volume of the RCSM, which partially offsets the shrinkage that occurs during concrete hardening. Through this mechanism, the drying shrinkage coefficient and temperature shrinkage coefficient of the RCSM can be effectively reduced. (3) Compared with the pure mixed mixture with 40% RA, adding 10% FA can reduce the shrinkage coefficient and average temperature shrinkage coefficient of the RCSM. However, if the FA content is insufficient or excessive, it will increase the shrinkage coefficient and average temperature shrinkage coefficient of the mixture.

From Figure 4c,f, it can be seen that by mixing 40% RA to prepare the RCSM and further adding AALS and FA, the dry shrinkage coefficient and average temperature shrinkage coefficient of the mixture can be significantly reduced. When the content of AALS and FA is 15% and 10%, respectively, the dry shrinkage coefficient and average temperature shrinkage coefficient of the composite-improved RCSM are 25.8% and 14.8% lower than those of the pure 40% RA mixture, and only a small number of fine cracks are seen on the surface of the specimen after the test.

By mixing AALS and FA into the RCSM, the shortcomings of single material improvement can be compensated for, and the composite improvement of mechanical properties and shrinkage performance of the mixture can be achieved. This is beneficial to promoting the safe service of the RCSM and improving its durability.

### 3.2. Optimization Design Based on Response Surface Methodology

To determine the optimal dosages of AALS, FA, and RA, the response surface method was used to optimize the design based on the previous test results. The dosages of AALS (A), FA (B), and RA (C) were taken as independent variables, while the 28-day unconfined compressive strength, 28-day splitting strength, 28-day flexural strength, 28-day compressive modulus of elasticity, 28-day drying shrinkage coefficient, and average temperature shrinkage coefficient of the composite-improved RCSM were considered as response variables. The Box–Behnken Design (BBD) method from the response surface method was employed to establish a model. The levels of response surface test factors and the experimental design table are presented in Tables 9 and 10.

**Table 9.** Response surface test factor levels.

| Factor | Code | Level | | |
|---|---|---|---|---|
| | | −1 | 0 | 1 |
| AALS/% | A | 10 | 15 | 20 |
| FA/% | B | 5 | 10 | 15 |
| RA/% | C | 25 | 40 | 55 |

**Table 10.** Response surface test design table.

| No. | A | B | C | 28 d Unconfined Compressive Strength (MPa) | 28 d Splitting Strength (MPa) | 28 d Flexural Strength (MPa) | 28 d Compressive Modulus of Elasticity (MPa) | 28 d Dry Shrinkage Coefficient ($10^{-6}$) | Average Temperature Shrinkage ($10^{-6}$/°C) |
|---|---|---|---|---|---|---|---|---|---|
| 1 | 20 | 5 | 40 | 7.11 | 0.55 | 1.43 | 1530.2 | 180.4 | 8.17 |
| 2 | 15 | 10 | 40 | 7.28 | 0.59 | 1.47 | 1583.2 | 168.5 | 7.89 |
| 3 | 10 | 10 | 55 | 7.02 | 0.53 | 1.36 | 1492.3 | 193.2 | 8.85 |
| 4 | 15 | 15 | 25 | 6.99 | 0.55 | 1.4 | 1471.2 | 177.8 | 7.61 |
| 5 | 15 | 10 | 40 | 7.26 | 0.58 | 1.48 | 1591.4 | 169.6 | 7.77 |
| 6 | 20 | 10 | 55 | 7.06 | 0.55 | 1.36 | 1501.8 | 187.3 | 8.74 |
| 7 | 15 | 10 | 40 | 7.26 | 0.59 | 1.49 | 1586.6 | 168.8 | 7.91 |
| 8 | 10 | 15 | 40 | 7.16 | 0.54 | 1.45 | 1536.1 | 184.1 | 8.28 |
| 9 | 20 | 15 | 40 | 7.16 | 0.55 | 1.45 | 1548.3 | 178.1 | 8.13 |
| 10 | 20 | 10 | 25 | 7.02 | 0.55 | 1.41 | 1468.5 | 179.2 | 7.64 |
| 11 | 15 | 5 | 25 | 6.91 | 0.53 | 1.37 | 1448.9 | 181.5 | 7.68 |
| 12 | 15 | 15 | 55 | 7.03 | 0.54 | 1.36 | 1495.8 | 185.6 | 8.68 |
| 13 | 15 | 10 | 40 | 7.27 | 0.58 | 1.47 | 1581.7 | 167.9 | 7.96 |
| 14 | 10 | 10 | 25 | 6.96 | 0.56 | 1.4 | 1458.6 | 185.8 | 7.85 |
| 15 | 10 | 5 | 40 | 7.01 | 0.53 | 1.41 | 1518.3 | 188.7 | 8.33 |
| 16 | 15 | 5 | 55 | 6.93 | 0.5 | 1.34 | 1482.9 | 189.3 | 8.71 |
| 17 | 15 | 10 | 40 | 7.25 | 0.59 | 1.48 | 1584.7 | 167.4 | 7.84 |

Using Design-Expert 10.0 software, two-dimensional contour plots and three-dimensional response surface plots were generated, as shown in Figures 5 and 6. Among them, the two-dimensional contour plot is the projection of the three-dimensional response surface plot onto the horizontal plane. The denser the contour lines and the steeper the fitted surface slope, the more significant the influence of that factor.

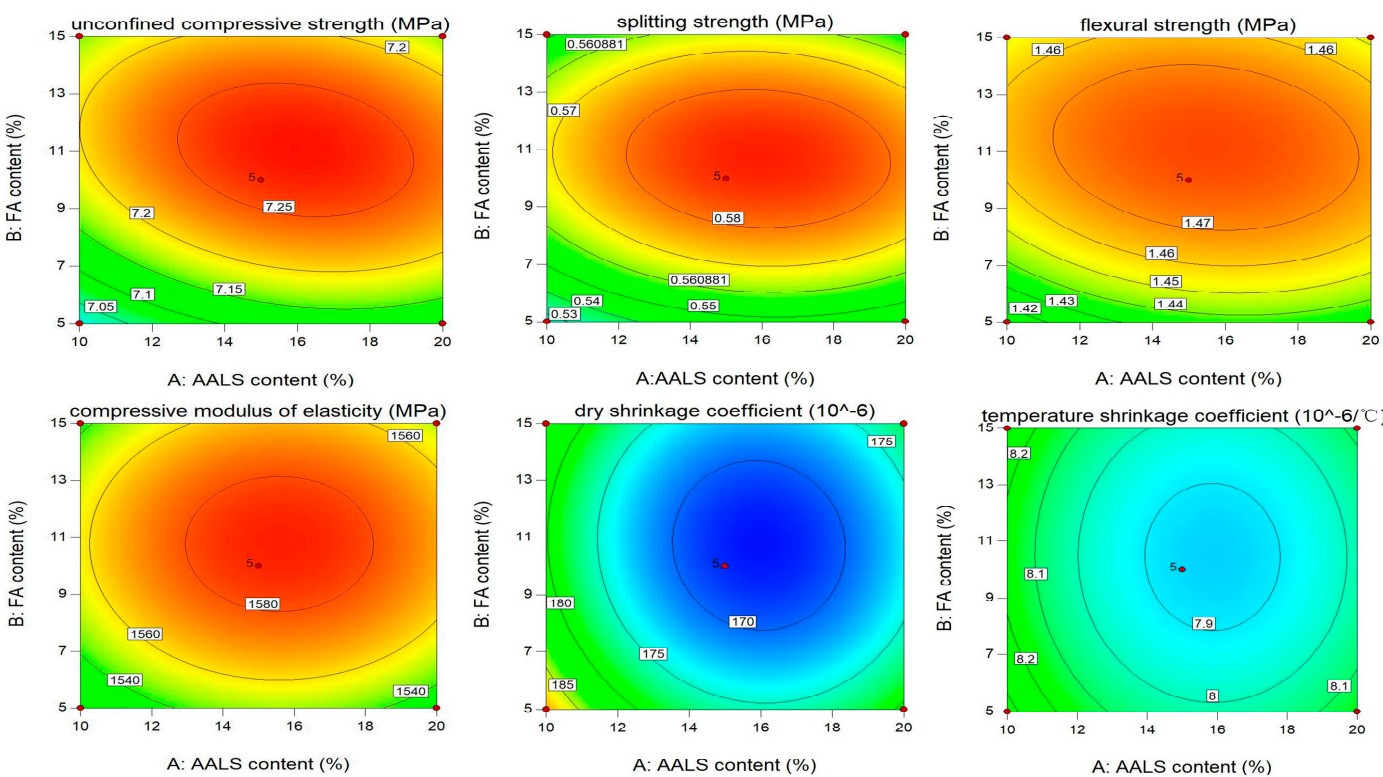

**Figure 5.** Two-dimensional contour plot between the independent variable and the response variable.

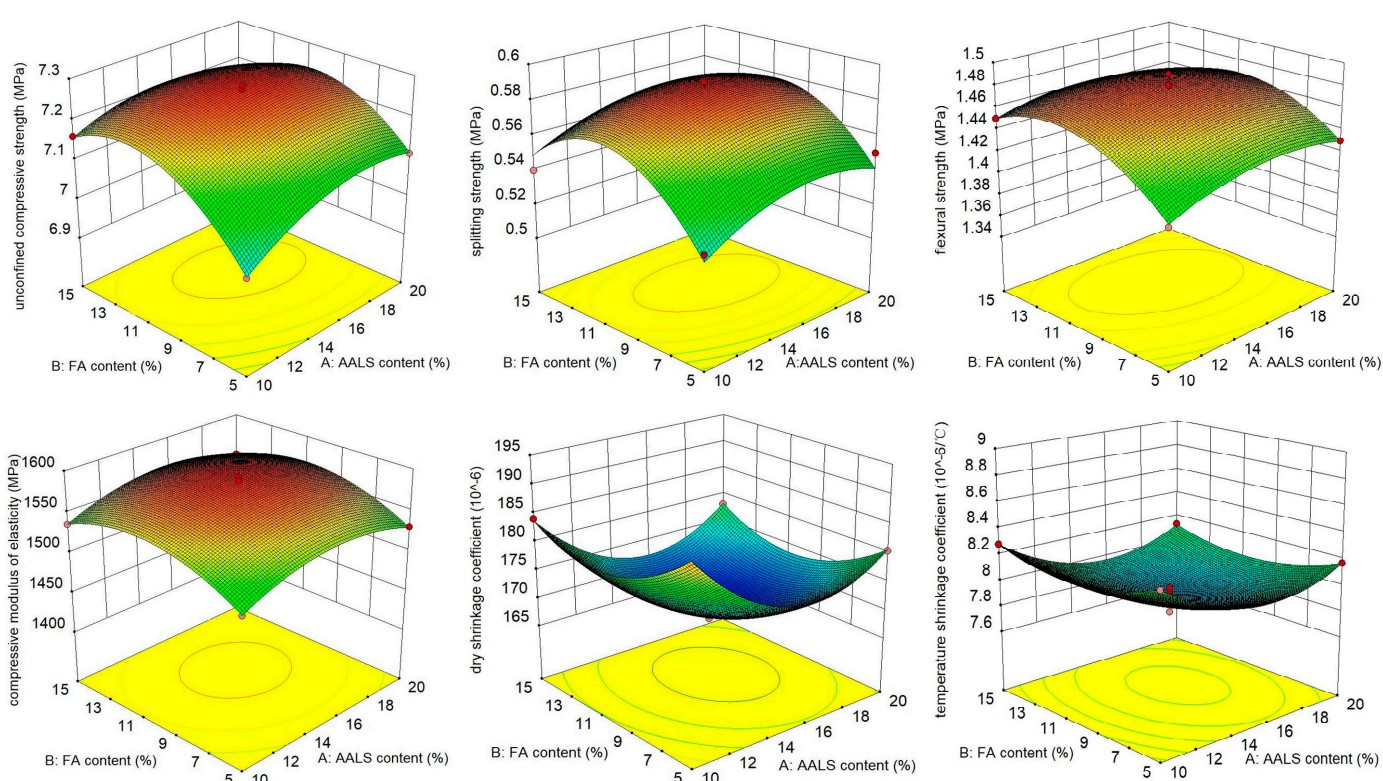

**Figure 6.** Three-dimensional surface plot between the independent variable and the response variable.

As can be seen from Figures 5 and 6, taking the AALS and FA additions as examples, the three-dimensional response surface slope of the independent variables and response variables is steep and the two-dimensional contour lines are dense, indicating that the

interaction of different additions of AALS, FA, and RA has a significant impact on the mechanical properties and durability of composite-improved RCSM. Taking the 28-day unconfined compressive strength as an example, a regression model between the independent variables and the response variable of the 28 d unconfined compressive strength was established as shown in Equation (1), and the variance calculation results of the regression model were calculated as shown in Table 11.

$$28 \text{ d unconfined compressive strength} = 7.26 + 0.025A + 0.048B + 0.02C - 0.025AB - 0.005AC + 0.005BC - 0.052A^2 - 0.1B^2 - 0.2C^2 \tag{1}$$

**Table 11.** ANOVA results of 28 d unconfined compressive strength indexes.

| Variance Source | Sum of Squared Deviations | Degrees of Freedom | Variance | F Value | p Value | Significance |
|---|---|---|---|---|---|---|
| Model | 0.27 | 9 | 0.03 | 268.29 | <0.0001 | Significant |
| A-Alkali-activated lithium slag | $5 \times 10^{-3}$ | 1 | $5 \times 10^{-3}$ | 45.45 | 0.0003 | |
| B-Fly ash | 0.018 | 1 | 0.018 | 164.09 | <0.0001 | |
| C-Recycled aggregates | $3.2 \times 10^{-3}$ | 1 | $3.2 \times 10^{-3}$ | 29.09 | 0.0010 | |
| AB | $2.5 \times 10^{-3}$ | 1 | $2.5 \times 10^{-3}$ | 22.73 | | |
| AC | $1 \times 10^{-4}$ | 1 | $1 \times 10^{-4}$ | 0.91 | | |
| BC | $1 \times 10^{-4}$ | 1 | $1 \times 10^{-4}$ | 0.91 | | |
| $A^2$ | 0.011 | 1 | 0.011 | 103.50 | <0.0001 | |
| $B^2$ | 0.044 | 1 | 0.044 | 398.24 | <0.0001 | |
| $C^2$ | 0.16 | 1 | 0.16 | 1485.51 | <0.0001 | |
| Residual | $7.7 \times 10^{-4}$ | 7 | $1.1 \times 10^{-4}$ | | | |
| Lack of Fit | $2.5 \times 10^{-4}$ | 3 | $8.3 \times 10^{-5}$ | 0.64 | 0.6276 | not significant |
| Pure Error | $5.2 \times 10^{-4}$ | 4 | $1.3 \times 10^{-4}$ | | | |
| Total | 0.27 | 16 | | | | |
| $R^2$ | 0.9971 | C.V. % | 0.15 | | | |

From Table 11, it can be seen that the *p* value of the model is less than 0.0001, which proves the feasibility of using response surface method to predict the 28 d unconfined compressive strength value. The *p* value of lack of fit for the model is 0.6276, indicating that the lack of fit caused by error is not significant. The correlation coefficient $R^2$ of the regression equation is 0.9971, indicating that 99.71% of the response variable results come from the selected independent variables, which proves the good correlation between the predicted and measured values. The CV value is only 0.15%, indicating that the regression model has high accuracy and that the total variation of the response variable cannot be explained by the mathematical model. To further illustrate the effectiveness of the mathematical model, the predicted values of 28 d unconfined compressive strength were compared with the actual values, as shown in Figure 7.

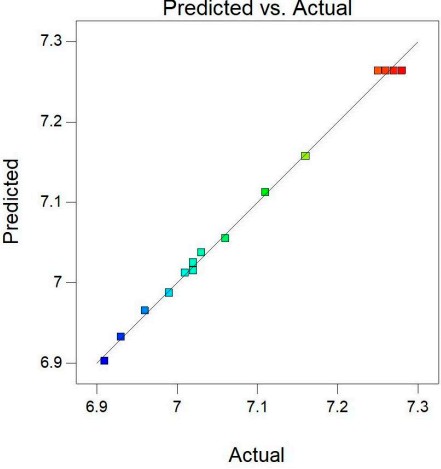

**Figure 7.** Comparison of actual and predicted values.

As shown in Figure 7, the 28 d unconfined compressive strength values are in good agreement with the model predictions, indicating that the selected model can accurately reflect the correlation between the response variable of the 28 d unconfined compressive strength and the independent variables of AALS/FA/RA. Similarly, the fitting equation between the 28 d splitting strength, 28 d flexural strength, 28 d compressive modulus of elasticity, 28 d dry shrinkage coefficient, and average temperature shrinkage coefficient values and the AALS/FA/RA content can be obtained, as shown in Table 12.

**Table 12.** Fitting equation for each test index.

| Test Indicators | Fitting Equation | $R^2$ |
|---|---|---|
| 28 d unconfined compressive strength | 28 d unconfined compressive strength $= 7.26 + 0.025A + 0.048B + 0.02C - 0.025AB - 0.005AC + 0.005BC - 0.052A^2 - 0.1B^2 - 0.2C^2$ | 0.9971 |
| 28 d splitting strength | 28 d splitting strength $= 0.59 + 0.005A + 0.00875B - 0.00875C - 0.0025AB + 0.0075AC + 0.005BC - 0.013A^2 - 0.03B^2 - 0.026C^2$ | 0.9523 |
| 28 d flexural strength | 28 d flexural strength $= 1.48 + 0.0038A + 0.014B - 0.02C - 0.005AB - 0.0025AC - 0.0025BC - 0.014A^2 - 0.029B^2 - 0.082C^2$ | 0.9911 |
| 28 d compressive modulus of elasticity | 28 d compressive modulus of elasticity $= 1585.52 + 5.444A + 8.89B + 15.7C + 0.075AB - 0.1AC - 2.35BC - 23.35A^2 - 28.95B^2 - 81.87C^2$ | 0.99.83 |
| 28 d dry shrinkage coefficient | 28 d dry shrinkage coefficient $= 168.44 - 3.35A - 1.79B + 3.89C + 0.57AB + 0.18AC + 8.6A^2 + 5.78B^2 + 9.33C^2$ | 0.9935 |
| average temperature shrinkage | average temperature shrinkage coefficient $= 7.87 - 0.079A - 0.024B + 0.52C + 0.0025AB + 0.025AC + 0.01BC + 0.23A^2 + 0.13B^2 + 0.17C^2$ | 0.9923 |

Based on the fitted equation in Table 12, the objective is to maximize the 28-day unconfined compressive strength, 28-day splitting strength, 28-day flexural strength, and 28-day compressive modulus of elasticity, while minimizing the 28-day drying shrinkage coefficient and average temperature shrinkage coefficient. Using the response surface method, the optimal dosages for the three admixtures were determined to be 15% AALS, 10% FA, and 40% RA. A comparison of the measured and predicted performance indicators for these admixture dosages is presented in Table 13. It can be observed that the optimal design determined using the response surface method achieves satisfactory prediction results.

**Table 13.** Properties of composite-improved RCSM at optimum dosage.

| Indicators | 28 d Unconfined Compressive Strength (MPa) | 28 d Splitting Strength (MPa) | 28 d Flexural Strength (MPa) | 28 d Compressive Modulus of Elasticity (MPa) | 28 d Dry Shrinkage Coefficient ($10^{-6}$) | Average Temperature Shrinkage ($10^{-6}/°C$) |
|---|---|---|---|---|---|---|
| projected value | 7.26 | 0.58 | 1.45 | 1585.5 | 167.6 | 7.68 |
| measured value | 7.29 | 0.59 | 1.48 | 1586.6 | 168.8 | 7.88 |

### 3.3. Microscopic Mechanism Analysis

#### 3.3.1. X-ray Diffraction

This article analyzes the internal mineral structure composition of a 40% recycled aggregate cement-stabilized macadam mixture with a curing period of 28 days and a composite-improved 40% recycled aggregate cement-stabilized macadam mixture with 15% AALS and 10% FA. The XRD patterns are shown in Figure 8. Among them, 2θ in the spectrum is the diffraction angle, which is the angle between the incident X-ray and the diffracted X-ray. It can accurately reflect the diffraction condition of X-rays in crystal materials and provide important information about the crystal structure, lattice constant, and phase composition of the material.

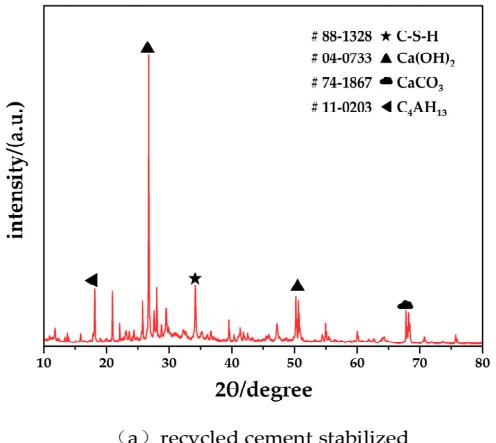

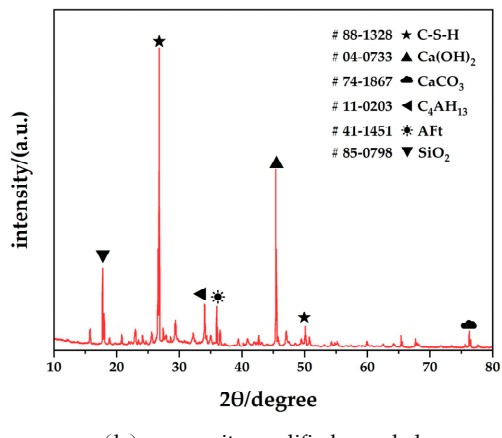

|（a）recycled cement stabilized macadam mixture | （b）composite modified recycled cement stabilized macadam mixture |

**Figure 8.** XRD patterns of RCSM and AALS–FA-composite-improved RCSM.

As shown in Figure 8, the main hydration products of RCSM are calcium hydroxide (C-H), hydrated silicate cementitious (C-S-H), hydrated tetracalcium aluminate ($C_4AH_{13}$), and a small amount of calcium carbonate $CaCO_3$). The main hydration products of the composite-improved RCSM are C-S-H, C-H, ettringite (AFt), $C_4AH_{13}$, $CaCO_3$, and unhydrated particles ($SiO_2$). The addition of AALS and FA causes a reaction with the cement hydration product C-H to form a large amount of C-S-H cementitious, consuming calcium hydroxide and increasing hydrated calcium silicate. Other studies have also obtained similar results [28,29]. In addition, Tan et al. [30] found that silicon dioxide and alumina in lithium slag can react with C-H to produce more C-S-H and AFt products. This further verifies the increased C-S-H peak and the appearance of the AFt peak in Figure 8b. In summary, the addition of AALS and FA significantly promoted the hydration reaction and generated more products such as C-S-H and AFt, which provided a solid strength basis for the mixture. At the same time, AFt has a micro-expansion effect [31], which can offset the shrinkage caused by the hardening process of some mixtures. These findings effectively confirm the previous analysis of the reasons for the improvement of mixture performance.

3.3.2. SEM Test

Using scanning electron microscopy, the microstructure and reaction process of the 40% recycled aggregate cement-stabilized macadam mixture with a curing period of 28 days and the 40% recycled aggregate cement-stabilized macadam mixture with a curing period of 28 days improved using 15% AALS and 10% FA were observed and analyzed at a magnification of 5000 times. The influence of AALS and FA on the RCSM was analyzed. The test results are shown in Figure 9.

As shown in Figure 9, The hydration products of the RCSM are not tightly wrapped and interwoven, showing many flakes of calcium hydroxide (C-H), accompanied by a small amount of flocculent and needle-like hydrated calcium silicate gel (C-S-H). A long crack with a length of approximately 80 μm can be seen in the SEM image, and several short cracks with lengths of approximately 20 μm are extended around it; there are also many micropores with diameters of approximately 5 μm. The microstructure of the composite-improved RCSM is dense, the hydration products and components are evenly distributed, and they are closely wrapped with each other, and only short cracks of approximately 10 μm are seen. This is because the AALS and FA participate in a secondary hydration reaction, producing more C-S-H and AFt [30]. The newly formed products fill the gaps, significantly reducing micropores and cracks, making the bond between the aggregate and the paste tighter and effectively improving the internal interface transition zone performance of the RCSM. This provides a good explanation for the improved strength and shrinkage performance of the RCSM after the addition of AALS and FA.

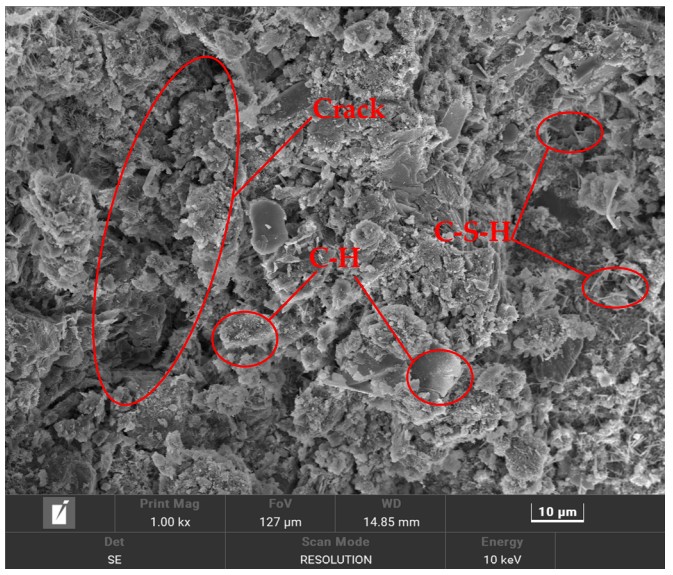

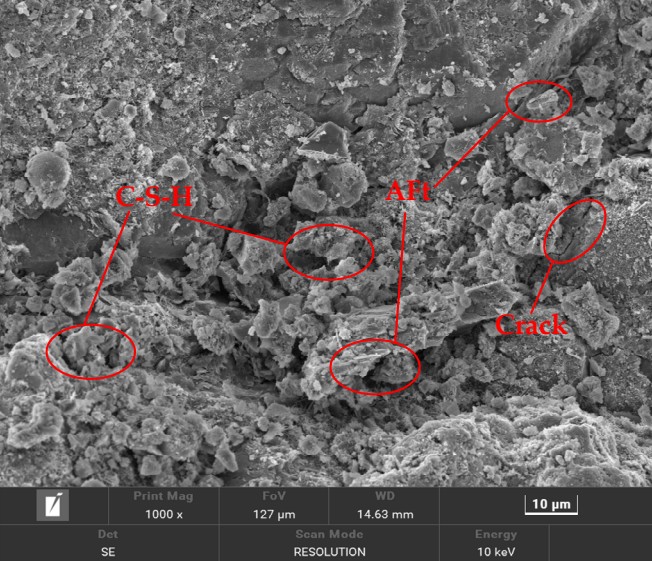

（a）recycled cement stabilized macadam mixture

（b）composite improved recycled cement stabilized macadam mixture

**Figure 9.** SEM images of RCSM and AALS–FA-composite-improved RCSM.

### 4. Conclusions

Through a series of systematic indoor tests, in this study, a thorough investigation was conducted into the impact of the admixture of AALS, FA, and RA on the mechanical properties and shrinkage performance of CSM. Using the response surface method, the optimal dosages of AALS, FA, and RA in the composite-improved RCSM were selected. Furthermore, the microstructural characteristics of the mixture were analyzed through XRD diffraction testing and SEM testing. The key findings are as follows:

(1) The AALS and FA were used to improve the RCSM, which significantly improved its mechanical properties and shrinkage properties. The incorporation of AALS can effectively improve the shrinkage performance of the mixture, and FA can improve the strength of the mixture more significantly.

(2) With the increase in RA content, the strength of RCSM increases first and then decreases. When the RA is less than 40%, the shrinkage performance of the mixture decreases slightly.

(3) The response surface method was used to optimize the design of the conventional performance test, and the optimum content of AALS, FA and RA in the composite-improved RCSM was optimized. The optimum content was AALS 15%, FA 10%, and RA 40%.

(4) Through microscopic experiments, it was found that the main hydration products of AALS–FA-composite-improved RCSM are C-S-H, C-H and AFt. AALS and FA play a pozzolanic activity. The generated C-S-H and AFt fill the gap, effectively improve the performance of the internal interface transition zone of the RCSM, and enhance the strength and shrinkage performance of the mixture.

(5) The composite-improved RCSM proposed in this paper not only makes full use of RA, but also further exerts the potential application value of lithium slag and fly ash, which provides technical support for the sustainable development of resource reuse and road construction. The next step will be a comprehensive assessment of the fatigue, frost resistance and erosion resistance of the mixture, and laying test sections in actual road projects. It is expected that under reasonable maintenance, the mixture will show sufficient durability and stability to meet long-term use needs. In addition, the cost will be reduced by optimizing the ratio, and attention will be paid to its maintenance needs, such as crack repair.

**Author Contributions:** Conceptualization, W.Y.; writing—original draft preparation, Z.J.; visualization, J.Y.; investigation, J.H. and X.H.; supervision, X.Y. and G.L.; software, C.W. All authors have read and agreed to the published version of the manuscript.

**Funding:** This research was funded by Hunan Provincial Water Resources Department Water Resources Science and Technology Project, XSKJ2022068-24.

**Data Availability Statement:** The data presented in this study are available on request from the corresponding author.

**Conflicts of Interest:** He Jiangang is an employee of Hunan Renjian Baogu High-Tech Development Co. The paper reflects the views of scientists, rather than the views of the company.

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
