# Peer review of "Study on the Performance of Recycled Cement-Stabilized Macadam Mixture Improved Using Alkali-Activated Lithium Slag–Fly Ash Composite"

_minerals, doi:10.3390/min14040418_

Round 1
Reviewer 1 Report (Previous Reviewer 1)
Comments and Suggestions for Authors
This study offers a valuable investigation into enhancing the mechanical properties and shrinkage performance of recycled cement stabilized macadam mixtures by incorporating alkali-activated lithium slag, fly ash, and recycled aggregate. The experimental design is systematic, covering key performance indicators and utilizing response surface methodology for optimization. The findings demonstrate significant improvements in compressive strength, splitting strength, and shrinkage resistance with the optimal admixture dosages. Microstructural analyses provide mechanistic insights into the enhancements. While the research is technically sound, a few aspects could be clarified or expanded, such as explaining some methodology details, providing additional context for the results, and further discussing practical implications and limitations. Nonetheless, this work makes a solid contribution to advancing sustainable road construction by presenting an effective approach to recycling waste materials in cement stabilized macadam.
- The abstract provides a good overview, but it would be helpful to quantify some of the key findings, such as the percentage improvements in strength and shrinkage resistance.
- The authors mention that "the performance of recycled aggregate is affected by various factors such as content and particle size." It would be informative to briefly explain how these factors influence performance.
- The research objectives and significance are clearly stated. However, the novelty of this work compared to previous studies could be articulated more explicitly. What are the key gaps in the existing literature that this study aims to address?
- The apparent relative density values for natural fine aggregate and recycled fine aggregate seem unusually low. Please double-check these values.
- The description of the mix design process is quite brief. Additional details on the rationale behind the selected cement content, aggregate gradation, and other mix parameters would be useful.
- How many replicates were tested for each performance indicator? Specifying the number of replicates would help assess the reliability of the results.
- The compressive strength and splitting strength plots are informative. It would be helpful to add error bars indicating the variability in the measured values.
- The authors attribute the strength improvements to the formation of calcium silicate hydrate and calcium aluminate hydrate. Were these phases directly observed in the microstructural analysis? If so, mentioning that here would strengthen the argument.
- The drying and temperature shrinkage results are interesting. Did the authors observe any visible cracking in the specimens at the end of the shrinkage tests? If so, how did the crack patterns differ between the mixtures?
- The use of response surface methodology is a nice aspect of this study. However, more details on the model fitting process and diagnostics would be appreciated. For example, what was the R-squared value for each fitted model?
- The fitted equations are useful for predicting performance based on admixture dosages. Have the authors considered providing a simplified nomogram or graphical tool to make these predictions more accessible to practitioners?
- The XRD analysis provides valuable insights into the hydration products. However, the discussion of these results is quite brief. Expanding the interpretation and linking it more closely to the observed performance improvements would be beneficial.
- The SEM images nicely illustrate the microstructural changes with admixture incorporation. Adding some quantitative analysis, such as measuring the pore size distribution or crack density, could further strengthen the interpretations.
- The conclusions effectively summarize the main findings. However, they could be expanded to discuss the broader implications of this work for sustainable construction and waste utilization.
- The authors mention plans for future durability testing and field validation, which is great. It would also be interesting to comment on the expected long-term performance and potential maintenance requirements of these recycled cement stabilized macadam mixtures.
- The reference list covers many relevant studies. However, it would be good to include a few more recent references (from the last 2-3 years) to demonstrate the current state-of-the-art in this field.
- The writing is generally clear, but there are a few grammatical errors and awkward phrases throughout the manuscript. A thorough proofreading and editing pass would enhance the readability.
- The figures and tables are informative, but some of them (e.g., Figures 2 and 3) are quite small and difficult to read. Increasing the font size and resolution would improve the visual clarity.
- The study focuses on the engineering properties of the recycled cement stabilized macadam mixtures, which is valuable. However, it would be interesting to also consider the environmental and economic aspects. For example, what are the potential carbon footprint reductions and cost savings associated with using these waste materials?
Comments on the Quality of English Language
Minor editing of English language required
Author Response
Detailed Response
We are very grateful for your constructive feedback. In order to further strengthen our research work, we have made a comprehensive revision of the manuscript and responded to each comment one by one.
Comment 1: The abstract provides a good overview, but it would be helpful to quantify some of the key findings, such as the percentage improvements in strength and shrinkage resistance.
Response 1: Thank you for your valuable comments on the abstract section. Regarding the suggestion that you mentioned to quantify the key findings, we mentioned in the abstract that the optimal dosage of alkali-activated lithium slag , fly ash, and recycled aggregate based on the response surface method is 15 %, 10 %, and 40 %, respectively, and listed in detail the performance improvement of the composite improved recycled cement stabilized macadam mixture compared with the cement stabilized macadam mixture with 40 % recycled aggregate. Specifically, the 28-day compressive strength and 28-day splitting strength increased by 26.8 % and 22.9 %, respectively, while the dry shrinkage coefficient and average temperature shrinkage coefficient decreased by 25.8 % and 14.8 %, respectively. These specific percentage data have clearly demonstrated the key findings of this study and are reflected in the abstract.
Comment 2: The authors mention that "the performance of recycled aggregate is affected by various factors such as content and particle size." It would be informative to briefly explain how these factors influence performance.
Response 2: Thank you very much for your review comments on the article. We originally intended to express that the dosage and particle size of recycled aggregates will have an impact on the performance of recycled cement stabilized macadam mixtures. I am very sorry for any inconvenience caused to you. We have made corresponding modifications in the article.
Firstly, in line 48 of the article, we made modifications to the original text. Due to the crushing process of recycled aggregate, it is subjected to rolling and crushing, resulting in many weak points under stress, resulting in poor homogeneity and uneven distribution of weak points. These issues will lead to a decrease in the strength of recycled cement stabilized macadam mixture, therefore it is necessary to strictly control the dosage, particle size and other parameters of recycled aggregates.
In line 195 of the article, it is mentioned how the content of recycled aggregate affects the strength of the mixture. When the content of alkali-activated lithium slag and fly ash remains unchanged, with the increase of recycled aggregate content, the strength of the recycled cement stabilized macadam mixture shows a trend of first increasing and then decreasing. This is because when the dosage is small, the unhydrated old cement mortar coated on the surface of the recycled aggregate undergoes hydration reaction when it encounters water, strengthening the internal bonding force of the mixture and increasing the strength of the mixture. However, as the dosage continues to increase, the strength of the recycled aggregate itself decreases more than the strength produced by the hydration reaction of water and mud, so the strength of the mixture gradually decreases. To improve this situation, we further enhance the strength of the recycled mixture by adding fly ash and alkali-activated lithium slag.
In line 235 of the article, it is mentioned that the content and particle size of recycled aggregate have an impact on the shrinkage performance of the mixture. When the content of alkali-activated lithium slag and fly ash remains constant, with the increase of recycled aggregate content, the dry shrinkage coefficient and average temperature shrinkage coefficient of recycled cement stabilized macadam mixture maintain a slow increase. When the content exceeds 40%, the dry shrinkage coefficient and average temperature shrinkage coefficient increase rapidly. The reason for this phenomenon is that there are many pores on the surface of recycled aggregate, with high water absorption rate and fast water loss, leading to an increase in the dry shrinkage coefficient and temperature shrinkage coefficient of the mixture. When the content of recycled aggregate exceeds 40%, the impact becomes more significant. Especially the water absorption rate of recycled fine aggregate is higher, which has a greater impact on the shrinkage performance of the mixture. Therefore, it is necessary to control the dosage of recycled aggregates with smaller particle sizes.
Comment 3: The research objectives and significance are clearly stated. However, the novelty of this work compared to previous studies could be articulated more explicitly. What are the key gaps in the existing literature that this study aims to address?
Response 3: Thank you very much for your review comments on the article. In line 96 of the article, we made relevant supplements.After in-depth analysis of the current research status, we have found that scholars have actively explored the performance of recycled cement stabilized macadam mixture. Although there has been attention to the improvement of the performance of recycled cement stabilized macadam mixtures by external additives, there is still insufficient research on the utilization of industrial solid waste, especially lithium slag and fly ash. In response to this situation, this article aims to make up for the shortcomings of existing research, deeply explore the impact of adding lithium slag and fly ash, two industrial solid wastes, on the mechanical properties and durability of recycled cement stabilized macadam mixtures, and analyze their mechanisms, in order to promote the green and sustainable process of road construction.
Comment 4: The apparent relative density values for natural fine aggregate and recycled fine aggregate seem unusually low. Please double-check these values.
Response 4: In response to the comments from the reviewers regarding the abnormally low apparent relative density values of natural and recycled fine aggregates, a careful examination and re measurement were conducted. The corrected data shows that the apparent relative density value of natural fine aggregate is 2.793g·cm-3, and the apparent relative density value of recycled fine aggregate is 2.569g·cm-3.
Comment 5: The description of the mix design process is quite brief. Additional details on the rationale behind the selected cement content, aggregate gradation, and other mix parameters would be useful.
Response 5: Thank you very much for your review comments on the article. The description of the mixed design process has been supplemented and refined based on your suggestions. In line 136 of the article, the basis for the selected parameters such as cement content and aggregate gradation was further explained. According to reference [25], in order to improve the mechanical properties of recycled cement stabilized macadam mixture, the cement mass fraction used in indoor experiments is 5%.
In order to meet the technical requirements of Chinese specification [26] for first-class highway, the C-B-3 median grading of skeleton dense grade structure is selected for this study. In line 145 of the article, the optimal moisture content and maximum dry density determined by the compaction test were added.
Meanwhile, in lines 147, 164, and 178 of the article, we have provided detailed supplementary explanations for Strength testing, shrinkage testing, and microscopic testing.
[25] Yang, S.; Yang, Z.; Chen, K.; Zhang, L.; Chu, J. Using case study on the performance of rural cement stabilizing construction waste recycling road base. Construction and Building Materials 2023, 405, 133351, doi:10.1016/j.conbuildmat.2023.133351.
[26] JTG/T F20-2015. In Technical Guidelines for Construction of Highway Roadbases; Ministry of Transport of the People’s Republic of China: Beijing, China, 2015.
Comment 6: How many replicates were tested for each performance indicator? Specifying the number of replicates would help assess the reliability of the results.
Response 6: Thank you for your valuable feedback on the article.In lines 156 and 170 of the article, we provided additional explanations for duplicate samples. Conduct 5 parallel tests on various mechanical properties and mix proportions. Dry shrinkage and temperature shrinkage tests, 3 parallel tests for each mix proportion.
Comment 7: The compressive strength and splitting strength plots are informative. It would be helpful to add error bars indicating the variability in the measured values.
Response 7: Thank you very much for your comments and valuable suggestions on the article. In lines 194 and 234 of the article, error bars are added in Figures 2 and 3 to indicate the variability of the measured values.
Comment 8: The authors attribute the strength improvements to the formation of calcium silicate hydrate and calcium aluminate hydrate. Were these phases directly observed in the microstructural analysis? If so, mentioning that here would strengthen the argument.
Response 8: Thank you very much for the valuable comments of the reviewers. We have made the corresponding supplement in the 210 lines of the article. More irregular petal-like calcium silicate hydrates and prismatic ettringite were observed in the microscopic experiments. The morphological and structural characteristics of these hydration products provide direct evidence to verify the above reaction and performance improvement process. The formation of hydrated calcium silicate and ettringite not only refines the voids of the mixture, optimizes the performance of the interfacial transition zone, but also improves the compactness of the overall structure, thus significantly enhancing the unconfined compressive strength and splitting strength of the recycled cement stabilized macadam mixture.
Comment 9: The drying and temperature shrinkage results are interesting. Did the authors observe any visible cracking in the specimens at the end of the shrinkage tests? If so, how did the crack patterns differ between the mixtures?
Response 9: Thank you very much for your valuable suggestions for the article, and give a detailed reply here.
After the shrinkage test, we did observe the specimens carefully, and found the existence of cracks on some specimens. These cracks are mainly caused by the shrinkage stress generated by the material during drying and temperature changes. In line 239 of the article, it is added that with the increase of recycled aggregate content, the shrinkage performance of the mixture gradually deteriorates, resulting in a larger number or wider cracks. In line 264 of the article, the shrinkage performance of the optimal mix proportion mixture is significantly enhanced, with only a small number of fine cracks visible.
Comment 10: The use of response surface methodology is a nice aspect of this study. However, more details on the model fitting process and diagnostics would be appreciated. For example, what was the R-squared value for each fitted model?
Response 10: Thank you very much for your valuable suggestions for the article. In line 304 of the article, we take the results of the analysis of variance of unconfined compressive strength as an example to show the relevant details of the model. In line 325 of the article, table 12, we show the fitting equation and R2 of each test index of the response surface method.
Comment 11: The fitted equations are useful for predicting performance based on admixture dosages. Have the authors considered providing a simplified nomogram or graphical tool to make these predictions more accessible to practitioners?
Response 11: Thank you very much for your valuable suggestions for the article. In line 317 of the article, we take the predicted value of 28 unconfined compressive strength value and the actual value as an example, and draw the comparison result diagram. The results show that the 28 d unconfined compressive strength value is in good agreement with the model prediction value, and the selected model can accurately reflect the correlation between the response variable of 28 d unconfined compressive strength value and the independent variable of alkali-activated lithium slag / fly ash / recycled aggregate.
In line 325 of the article, the fitting equation of each test index is further demonstrated. Finally, in line 335 of the article, the predicted and measured values of each index of the mixture under the optimal dosage are displayed in detail in the form of tables. Through comparative analysis, it is found that the optimal design of the optimal dosage of admixture based on the response surface method can achieve ideal prediction results.
Comment 12: The XRD analysis provides valuable insights into the hydration products. However, the discussion of these results is quite brief. Expanding the interpretation and linking it more closely to the observed performance improvements would be beneficial.
Response 12: Thank you very much for your review comments, we expanded the interpretation of the XRD test and linked it more closely to the observed performance improvement. In line 357 of the article, it was added that the addition of alkali-activated lithium slag and fly ash significantly promoted the hydration reaction and generated more products such as C-S-H and AFt. These hydration products provide a solid strength foundation for the mixture. At the same time, it can be seen from the literature [31] that AFt has a micro-expansion effect, which can offset the shrinkage caused by the hardening process of some mixtures. These findings well confirm the previous analysis of the reasons for the improvement of mixture performance.
[31]Wang, L.; Shen, A.; Lyu, Z.; Guo, Y.; He, Z.; Mou, G.; Wei, Z. Rapid regeneration cement-stabilized macadam: Preparation, mechanical properties, and dry shrinkage performance. Construction and Building Materials 2022, 341, 127901, doi:10.1016/j.conbuildmat.2022.127901.
Comment 13: The SEM images nicely illustrate the microstructural changes with admixture incorporation. Adding some quantitative analysis, such as measuring the pore size distribution or crack density, could further strengthen the interpretations.
Response 13: We are very grateful for the valuable suggestions put forward by the reviewer. As supplemented in lines 374 and 379 of the article, a long crack about 80μm in length can be seen in the SEM image of the recycled cement stabilized macadam mixture, with multiple short cracks about 20μm in length extending around it. There are also many pores with a diameter of about 5μm. However, in the composite modified recycled cement stabilized macadam mixture, alkali-activated lithium slag and fly ash participate in the secondary hydration reaction, generating more irregular petal-like C-S-H and prismatic AFt. The SEM image shows that the newly generated substances fill the gaps, greatly reducing the pores and gaps. Only short cracks about 10μm in length are visible, effectively improving the microstructure of the mixture.
Comment 14: The conclusions effectively summarize the main findings. However, they could be expanded to discuss the broader implications of this work for sustainable construction and waste utilization.
Response 14: In the 409th line of the article, we further expand the conclusion and show that the composite improved recycled cement stabilized macadam mixture proposed in this paper not only makes full use of recycled aggregate, but also excavates the potential value of lithium slag and fly ash, which provides technical support for the realization of resource reuse and sustainable development of road construction.
Comment 15: The authors mention plans for future durability testing and field validation, which is great. It would also be interesting to comment on the expected long-term performance and potential maintenance requirements of these recycled cement stabilized macadam mixtures.
Response 15: Thank you very much for the expert opinion, in the article line 414 we have carried on the related supplement. The next step will be a comprehensive assessment of the fatigue, frost resistance and erosion resistance of the mixture, and laying test sections in actual road projects. It is expected that under reasonable maintenance, the mixture will show excellent durability and stability to meet long-term use needs. In addition, it will also pay attention to its maintenance needs, such as crack repair.
Comment 16: The reference list covers many relevant studies. However, it would be good to include a few more recent references (from the last 2-3 years) to demonstrate the current state-of-the-art in this field.
Response 16: Thank you very much for your review comments. We have collated and supplemented the references, and added the relevant literatures published in the past 2-3 years. These literatures cover the latest research results, methods and viewpoints, which can better support and enrich our research work. The specific new references are :
[1]Yan, P.; Ma, Z.; Li, H.; Gong, P.; Liu, Z.; Han, J.; Xu, M.; Hua, S. Evaluation of the shrinkage properties and crack resistance performance of cement-stabilized pure coal-based solid wastes as pavement base materials. Construction and Building Materials 2024, 421, 135680, doi:10.1016/j.conbuildmat.2024.135680.
[2]Chiranjeevi, K.; R G, Y.; Kumar, D.H.; Mulangi, R.H.; Ravi Shankar, A.U. Utilization of recycled concrete aggregates for pavement base courses – A detailed laboratory study. Construction and Building Materials 2024, 411, 134122, doi:10.1016/j.conbuildmat.2023.134122.
[5]Gu, Z.; Zhang, Y.; Luo, X.; Li, H.; Liu, G. Systematical calibration and validation of discrete element models for fiber reinforced cement treated aggregates. Construction and Building Materials 2023, 392, 131832, doi:10.1016/j.conbuildmat.2023.131832.
[6]Tang, Q.; Tian, A.; Ling, C.; Huang, Y.; Gu, F. Physical and mechanical properties of recycled aggregates modified by microbially induced calcium carbonate precipitation. Journal of Cleaner Production 2023, 382, 135409, doi:10.1016/j.jclepro.2022.135409.
[25]Deng, C.; Jiang, Y.; Tian, T.; Yi, Y. Laboratory mechanical properties and frost resistance of vibration-compacted cement–fly ash slurry and cement–fly ash-treated macadam mixtures. Construction and Building Materials 2024, 419, 135555, doi:10.1016/j.conbuildmat.2024.135555
[31]Wang, L.; Shen, A.; Lyu, Z.; Guo, Y.; He, Z.; Mou, G.; Wei, Z. Rapid regeneration cement-stabilized macadam: Preparation, mechanical properties, and dry shrinkage performance. Construction and Building Materials 2022, 341, 127901, doi:10.1016/j.conbuildmat.2022.127901.
Comment 17: The writing is generally clear, but there are a few grammatical errors and awkward phrases throughout the manuscript. A thorough proofreading and editing pass would enhance the readability.
Response 17: Thank you very much for the expert's review comments. We sincerely apologize for the grammatical errors and unsmooth expressions you mentioned, and we attach great importance to them. We have conducted a comprehensive review and revision of the paper, focusing on grammatical errors, expression fluency, and logical coherence, to ensure that the paper is more accurate, clear, and fluent in language expression. Thank you again for your review comments and valuable suggestions.
Comment 18: The figures and tables are informative, but some of them (e.g., Figures 2 and 3) are quite small and difficult to read. Increasing the font size and resolution would improve the visual clarity.
Response 18: Thank you very much for your valuable comments. In lines 194 and 234 of the article, Figure 2 and Figure 3 have been carefully adjusted to significantly enlarge the font size in the chart and further improve the resolution to ensure the clarity and readability of the chart information. Thanks again to the reviewers for their careful guidance and advice.
Comment 19: The study focuses on the engineering properties of the recycled cement stabilized macadam mixtures, which is valuable. However, it would be interesting to also consider the environmental and economic aspects. For example, what are the potential carbon footprint reductions and cost savings associated with using these waste materials?
Response 19: Thank you very much for your valuable comments. In the 37th line of the article, it is mentioned that if the waste concrete slab is directly discarded, it will occupy the land resources, cause serious waste of resources, and bring unnecessary economic losses due to the failure to make full use of these abandoned slabs. At the same time, the chemical substances contained in waste concrete may gradually infiltrate into soil and groundwater, posing a long-term and irreversible pollution risk to the ecological environment.
Therefore, in line 44, it is mentioned that the recycled aggregate is prepared by using the waste concrete plate, and then the recycled cement stabilized macadam mixture is prepared by using the recycled aggregate. This practice not only promotes the recycling of waste mineral resources, but also reduces the pollution of construction waste to the environment, and achieves the dual goals of resource recycling and environmental protection.
At the same time, in the last line 415 of the article, the next task will be to pave the test section and further optimize the ratio to reduce the project cost.

Reviewer 2 Report (Previous Reviewer 2)
Comments and Suggestions for Authors
I think this paper can be published in the Minerals journal, because it has been revised in detail according to the reviewer comments and meets the relevant requirements of the Minerals journal
Author Response
Detailed Response
We are very grateful for your constructive feedback. In order to further strengthen our research work, we have made a comprehensive revision of the manuscript and responded to each comment one by one.
Comment 1: I think this paper can be published in the Minerals journal, because it has been revised in detail according to the reviewer comments and meets the relevant requirements of the Minerals journal.
Response 1: Thank you very much for the reviewer 's affirmation and opinion ! We sincerely thank the reviewers for their professional guidance and valuable suggestions during the review process. These opinions play a vital role in the perfection and improvement of the paper.

Reviewer 3 Report (New Reviewer)
Comments and Suggestions for Authors
Article reports an experimental work on the effects of additions on performance of recycled cement stabilized macadam mixture. It is an interesting topic that complement some existing works in this subject and that were referenced by the authors in the state of the art.
In general the information is clear and coherent. However, in some moments there could be more detail in the procedures and results obtained. So, I present some comments and suggestions for improvement below:
- The importance of this study for science is not clear in the abstract. Abstract should better clarify the central focus of the paper. Abstract needs to be clearer;
- In introduction, authors present a resumed state of the art with the improvement or the decrease in the properties of recycled cement stabilized macadam base, analyzed by other authors. However, data analysis is reduced. The authors only give an up or down indication, sometimes mentioning some percentages of these evaluation, but the analysis needs a deeper justification for these variations. If authors have this data, it would be interesting to improve the state of the art with this information and discussion;
- The use of more acronyms should be considered, making reading easier. Alkali-activated lithium slag; recycled aggregates; recycled cement stabilized macadam; among others, are excessively used extensively;
- Standards used in test methods should be specified in references;
- More information about mix design of the specimens is important;
- Page 3, Figure 1: Comment: The “technical route” should be better explained previously. It appears out of place and the reader does not understand the details of the intended experimental work. To review.
- Page 3, line 115: “…all indicators meet the specifications”. Comment: Several times throughout the article, checking specifications is mentioned, but which ones? To review.
- Page 4, 2.1. Materials: Comment: Chemical composition of cement and more data about origin concrete of recycled aggregate should/could be mentioned. It can help on results discussion. To review if data exists;
- Page 5, Figure 2: Comment: The scale of the figures must be the same. This is the only way for the reader to compare data, since they do not have access to real values. To review all figures.
- Conclusions should be more objective and less descriptive. Some data from the conclusions should be in the discussion and some are repeated, becoming redundant.
- It is not clear what the major contributions of this manuscript are. Abstract and conclusions should be improved to ensure this lack.
Author Response
Detailed Response
We are very grateful for your constructive feedback. In order to further strengthen our research work, we have made a comprehensive revision of the manuscript and responded to each comment one by one.
Comment 1: The importance of this study for science is not clear in the abstract. Abstract should better clarify the central focus of the paper. Abstract needs to be clearer.
Response 1: In view of your suggestions on the abstract part, corresponding modifications and improvements have been made. In the abstract, lines 13 and 30 of the article, we clearly expounded the central focus of this study, that is, to fully tap the potential utilization value of lithium slag and fly ash, two kinds of industrial solid waste, in recycled cement stabilized macadam mixture, and to clarify the effect of adding lithium slag, fly ash and recycled aggregate on the mechanical properties and durability of cement stabilized macadam mixture. This study provides technical support for the sustainable development of resource reuse and road construction.
Comment 2: In introduction, authors present a resumed state of the art with the improvement or the decrease in the properties of recycled cement stabilized macadam base, analyzed by other authors. However, data analysis is reduced. The authors only give an up or down indication, sometimes mentioning some percentages of these evaluation, but the analysis needs a deeper justification for these variations. If authors have this data, it would be interesting to improve the state of the art with this information and discussion.
Response 2: Thank you very much to the reviewers for their valuable feedback on this article. We have made modifications and supplements to address the issue of insufficient data analysis in the introduction section that you mentioned. Specifically:
Lan et al.'s research further explains the effect of hydration reaction on the strength and rebound modulus of the base layer of unhydrated cement mortar on the surface of recycled aggregate, as well as the phenomenon that its own strength decreases more than the strength provided by hydration reaction with the increase of recycled aggregate content.
Yan et al. found that the high porosity and high water absorption characteristics of recycled aggregates lead to an increase in the dry shrinkage coefficient of the mixture.
Deng et al.'s discovery further explains that adding fly ash to cement stabilized macadam mixture can effectively improve the internal structure of the mixture and enhance its mechanical properties through its micro aggregate filling effect.
Zhang et al.'s study explained that the addition of alkali activated steel slag promotes hydration reaction, generates additional cementitious products, and significantly improves the compressive strength, frost resistance, and dry shrinkage performance of the mixture.
The study by He et al. explained that lithium slag undergoes secondary hydration reactions with cement hydration products, resulting in the formation of hydrated silicates and ettringite. These hydration products can effectively fill the pores inside the concrete, thereby improving the later strength of the concrete. And through single factor experiments, it was found that the optimal dosage of lithium slag in concrete is 20%.
Qin et al. found that adding 20% lithium slag and 30% recycled coarse aggregate to concrete increased the cube compressive strength and elastic modulus by 9.9% and 9.94%, respectively. The reason is that the addition of lithium slag promotes the development of concrete cementitious structure, improves its microstructure, and refines pores.
Comment 3: The use of more acronyms should be considered, making reading easier. Alkali-activated lithium slag; recycled aggregates; recycled cement stabilized macadam; among others, are excessively used extensively.
Response 3: Thank you very much for your review comments. We have defined the following abbreviations in the paper and replaced them in appropriate places:
Recycled cement stabilized macadam mixture (RCSM)
Cement stabilized macadam mixture (CSM)
Recycled aggregate (RA)
Alkali-activated lithium slag (AALS)
Fly ash (FA)
Comment 4: Standards used in test methods should be specified in references.
Response 4: According to your suggestion, the standards used for the testing method have been indicated in the references. The specific reference standards are as follows:
[24] JTG 3432-2024. In Test Methods of Aggregates for Highway Engineering; Ministry of Transport of the People’s Republic of China: Beijing, China, 2024.
[26] JTG/T F20-2015. In Technical Guidelines for Construction of Highway Roadbases; Ministry of Transport of the People’s Republic of China: Beijing, China, 2015.
[27] JTG E51-2009. In Test Methods of Materials Stabilized with Inorganic Binders for Highway Engineering; Ministry of Transport of the People’s Republic of China: Beijing, China, 2009.
Comment 5: More information about mix design of the specimens is important.
Response 5: Thank you very much for your review comments on the article. The description of the mixed design process has been supplemented and refined based on your suggestions. Firstly, in line 136 of the article, according to relevant literature, in order to improve the mechanical properties of recycled cement stabilized macadam mixture, the cement mass fraction used in indoor experiments is 5%. Secondly, in order to meet the technical requirements of Chinese specifications for first-class highway, the C-B-3 median grading of skeleton dense grade structure is selected for this study. In line 145 of the article, we added the optimal moisture content and maximum dry density determined by the compaction test. In line 147, 164, 178 of the article, relevant details of mechanical testing, shrinkage testing, and microscopic testing were added.
Comment 6: Page 3, Figure 1: Comment: The “technical route” should be better explained previously. It appears out of place and the reader does not understand the details of the intended experimental work. To review.
Response 6: Thank you very much for your valuable review comments. After careful review, we realized that the placement of Figure 1 was indeed abrupt and did not fully explain the details of the technical route. Considering this, in order to avoid duplication of content and confusion for readers, we have removed Figure 1. In lines 147, 164, and 178 of the article, we have provided detailed supplementary explanations for Strength testing, shrinkage testing, and microscopic testing.
Comment 7:Page 3, line 115: “…all indicators meet the specifications”. Comment: Several times throughout the article, checking specifications is mentioned, but which ones? To review.
Response 7: According to your feedback, corresponding specifications have been added to the references. Cite at lines 119 and 122 (JTG 3432-2024), line 138 (JTG/T F20-2015), and lines 142, 148, and 165 (JTG E51-2009) in the article. The specific reference specifications are as follows:
[24] JTG 3432-2024. In Test Methods of Aggregates for Highway Engineering; Ministry of Transport of the People’s Republic of China: Beijing, China, 2024.
[26] JTG/T F20-2015. In Technical Guidelines for Construction of Highway Roadbases; Ministry of Transport of the People’s Republic of China: Beijing, China, 2015.
[27] JTG E51-2009. In Test Methods of Materials Stabilized with Inorganic Binders for Highway Engineering; Ministry of Transport of the People’s Republic of China: Beijing, China, 2009.
Comment 8: Page 4, 2.1. Materials: Comment: Chemical composition of cement and more data about origin concrete of recycled aggregate should/could be mentioned. It can help on results discussion. To review if data exists.
Response 8: Thank you very much for your valuable review comments. In line 130 of the text, we added the chemical composition of cement. In line 120, we added information on the sources of recycled aggregates, indicating that these aggregates were obtained by crushing and processing the abandoned concrete pavement in Xiangtan, Hunan. Meanwhile, we provide detailed road performance indicators of recycled aggregates in lines 131 and 132 of the article.
Comment 9: Page 5, Figure 2: Comment: The scale of the figures must be the same. This is the only way for the reader to compare data, since they do not have access to real values. To review all figures.
Response 9: Thank you very much for the valuable comments provided by the reviewer. In line 194 of the article, we have redrawn the original Figure 2, with a focus on correcting the scale issue of the figures to maintain consistency in the scales of each figure. We have also made modifications to other images in the article that have similar issues. Secondly, we further enlarged the font size in the chart, improved resolution, and added error bars to ensure the readability and accuracy of the chart information. Thank you again for the careful guidance and suggestions from the reviewer.
Comment 10: Conclusions should be more objective and less descriptive. Some data from the conclusions should be in the discussion and some are repeated, becoming redundant.
Response 10: Thank you very much for the valuable feedback from the reviewers. We have made revisions to the conclusion section according to your request. The revised conclusion has removed descriptive content and duplicate content. Conclusion after correction:
(1) The alkali-activated lithium slag and fly ash were used to improve the recycled cement stabilized macadam mixture, which significantly improved its mechanical properties and shrinkage properties. The incorporation of alkali-activated lithium slag can effectively improve the shrinkage performance of the mixture, and fly ash can improve the strength of the mixture more significantly.
(2) With the increase of recycled aggregates content, the strength of recycled cement stabilized macadam mixture increases first and then decreases. When the recycled aggregates is less than 40 %, the shrinkage performance of the mixture decreases slightly.
(3) The response surface method was used to optimize the design of the conventional performance test, and the optimum content of alkali-activated lithium slag, fly ash and recycled aggregates in the composite improved recycled cement stabilized macadam mixture was optimized. The optimum content was alkali-activated lithium slag 15 %, fly ash 10 %, and recycled aggregates 40 %.
(4) Through microscopic experiments, it is found that the main hydration products of alkali-activated lithium slag-fly ash composite improved recycled cement stabilized macadam mixture are C-S-H, C-H and AFt. alkali-activated lithium slag and fly ash play a pozzolanic activity. The generated C-S-H and AFt fill the gap, effectively improve the performance of the internal interface transition zone of the recycled cement stabilized macadam mixture, and improve the strength and shrinkage performance of the mixture.
Comment 11: It is not clear what the major contributions of this manuscript are. Abstract and conclusions should be improved to ensure this lack.
Response 11: We have made modifications to the abstract section to further clarify the main contribution of this article. To comprehensively explore the potential utilization value of lithium slag and fly ash, two industrial solid wastes, in recycled cement stabilized macadam mixtures, and to clarify the impact of adding lithium slag, fly ash, and recycled aggregates on the mechanical properties and durability of cement stabilized macadam mixtures. By carrying out the relevant work in this article, the influence of adding alkali-activated lithium slag, fly ash, and recycled aggregate on the performance of cement stabilized macadam mixture was obtained, and the optimal dosage was determined. The mechanism of improving the performance of the mixture was analyzed through microscopic experiments.
The conclusion section has been revised to further clarify the main contribution of this article. The conclusion drawn from this article is that the composite improvement of recycled cement stabilized macadam mixture using alkali-activated lithium slag and fly ash significantly improves its mechanical and shrinkage properties. The specific effects of adding alkali-activated lithium slag, fly ash, and recycled aggregate on the performance of cement stabilized macadam mixtures were obtained, and the response surface method was used to determine the optimal dosage of alkali-activated lithium slag, fly ash, and recycled aggregate as 15%, 10%, and 40%, respectively. Finally, through microscopic experiments, the microscopic mechanism for improving the performance of the mixture was discovered. The composite improved recycled cement stabilized macadam mixture proposed in this paper not only makes full use of recycled aggregate, but also excavates the potential value of lithium slag and fly ash, which provides technical support for the sustainable development of resource reuse and road construction.

Reviewer 4 Report (New Reviewer)
Comments and Suggestions for Authors
After reading the revised version and considering the inclusion of news information, I consider that the manuscript has been substantially improved. However, there are two small points left, in the Introduction chapter, which I recommend the author’s review:
On page 2, lines 84 to 86 it is written: “Qin et 84 al. [22,23] believed that the addition of lithium slag significantly improves the compressive strength, tensile strength, and ductility of recycled concrete.” Why is it a matter of belief?
I think the novelty of the study is the use of lithium slag. This fact can be highlighted in the text on lines 86 to 93 on page 2.
After clarifying these points, I think the manuscript could be accepted for publication.
Author Response
Detailed Response
We are very grateful for your constructive feedback. In order to further strengthen our research work, we have made a comprehensive revision of the manuscript and responded to each comment one by one.
Comment 1: On page 2, lines 84 to 86 it is written: “Qin et 84 al. [22,23] believed that the addition of lithium slag significantly improves the compressive strength, tensile strength, and ductility of recycled concrete.” Why is it a matter of belief?
Response 1: Thank you for your careful correction. In the original description, there were indeed issues with inaccurate and specific expression, and we have made modifications and additions to this.
In line 87 of the article, we provide a detailed description of the experimental results of He et al. and Qin et al. Through indoor experiments, they found that adding lithium slag to concrete can significantly improve its strength and elastic modulus. Specifically, He et al. found through single factor experiments that the addition of lithium slag can effectively improve the later strength of concrete, and determined that the optimal amount of lithium slag is 20%. However, Qin et al. found that adding 20% lithium slag and 30% recycled coarse aggregate to concrete increased its cubic compressive strength and elastic modulus by 9.9% and 9.94%, respectively. The addition of lithium slag promoted the development of concrete cementitious structure, improved its microstructure, refined its pores, and significantly improved the overall performance of concrete.
Comment 2: I think the novelty of the study is the use of lithium slag. This fact can be highlighted in the text on lines 86 to 93 on page 2.
Response 2: Thank you very much to the reviewers for their careful review and valuable feedback on this article. In line 96 of the article, modifications and supplements were made, and scholars actively explored the performance of recycled cement stabilized macadam. Although there have been studies that have focused on the improvement of the performance of recycled cement stabilized macadam mixtures by external additives, there is still insufficient research on the utilization of industrial solid waste, especially lithium slag and fly ash. In response to this situation, this article aims to make up for the shortcomings of existing research, deeply explore the impact of adding lithium slag and fly ash, two industrial solid wastes, on the mechanical properties and durability of recycled cement stabilized macadam mixtures, and analyze their mechanisms, in order to promote the green and sustainable process of road construction.

Round 2
Reviewer 1 Report (Previous Reviewer 1)
Comments and Suggestions for Authors
Well revised. Can be processed for the next stage of publication.
Author Response
Detailed Response
We are very grateful for your constructive feedback. In order to further strengthen our research work, we have made a comprehensive revision of the manuscript and responded to each comment one by one.
Comment 1: Well revised. Can be processed for the next stage of publication.
Response 1: Thank you very much for the reviewer 's affirmation and opinion ! We sincerely thank the reviewers for their professional guidance and valuable suggestions during the review process.

Reviewer 3 Report (New Reviewer)
Comments and Suggestions for Authors
Authors took into account most of the comments and suggestions made. Some more minor changes should be considered to improve the manuscript. Suggestions are below.
Minor changes:
- Abstract, Page 1, line 13: The changed sentence in line 13 still doesn't make sense. It can be an introduction to the topic, but it must have a logical sequence. This sentence should be reviewed as it is the first approach to the subject.
- Page 9, line 241: What do the authors mean with “And the specimens after the test were observed”?
- Conclusions, line 407: Review sentence. For instance, “optimized” is used twice in the same sentence.
- Conclusions, line 417: Review sentence. What do authors mean with “excavates”? In some specific moments throughout the all article, less technical language is used, as is the case here. This should be revised in the final version.
- Conclusions, line 421: Review sentence. How can the authors quantify/justify the "excellent durability". Consider changing to "adequate durability" or "similar/greater durability than...".
Author Response
Detailed Response
We are very grateful for your constructive feedback. In order to further strengthen our research work, we have made a comprehensive revision of the manuscript and responded to each comment one by one.
Comment 1: - Abstract, Page 1, line 13: The changed sentence in line 13 still doesn't make sense. It can be an introduction to the topic, but it must have a logical sequence. This sentence should be reviewed as it is the first approach to the subject.
Response 1: Thank you very much for your comments, in line 13 of the article was modified. The huge demand for sand and gravel resources in road engineering construction leads to excessive consumption of resources and environmental damage. Recycling waste concrete and industrial solid waste as a road material is a promising alternative. In order to explore the application of these solid wastes in the road base, this paper studies the effect of adding lithium slag activated by alkaline activator, adding fly ash and the combination of the two on the compressive strength, splitting strength and shrinkage performance of recycled cement stabilized macadam mixture.
Comment 2: - Page 9, line 241: What do the authors mean with “And the specimens after the test were observed”?
Response 2: Thank you very much for your comments, in line 240 of the article was modified. After the test, it was observed that with the increase of recycled aggregate content, the number and width of cracks on the surface of the sample gradually increased.
Comment 3: - Conclusions, line 407: Review sentence. For instance, “optimized” is used twice in the same sentence.
Response 3: Thank you very much for your comments, in line 408 of the article was modified. The generated C-S-H and AFt fill the gap, effectively improve the performance of the internal interface transition zone of the recycled cement stabilized macadam mixture, and enhance the strength and shrinkage performance of the mixture.
Comment 4: - Conclusions, line 417: Review sentence. What do authors mean with “excavates”? In some specific moments throughout the all article, less technical language is used, as is the case here. This should be revised in the final version.
Response 4: Thank you very much for your comments, we have revised and supplemented the full text, in line 411 of the article was modified. The modified expression is that the composite improved recycled cement stabilized macadam mixture proposed in this paper not only makes full use of recycled aggregate, but also further exerts the potential application value of lithium slag and fly ash, which provides technical support for the sustainable development of resource reuse and road construction.
Comment 5: - Conclusions, line 421: Review sentence. How can the authors quantify/justify the "excellent durability". Consider changing to "adequate durability" or "similar/greater durability than...".
Response 5: Thank you very much for your comments, in line 415 of the article was modified. The modified expression is that it is expected that under reasonable maintenance, the mixture will show sufficient durability and stability to meet long-term use needs.

This manuscript is a resubmission of an earlier submission. The following is a list of the peer review reports and author responses from that submission.
Round 1
Reviewer 1 Report
Comments and Suggestions for Authors
This is an interesting study investigating the effects of alkali-activated lithium slag and fly ash on improving the performance of recycled cement stabilized macadam mixtures. The paper is well-structured and easy to follow. The test methods seem appropriate. The results demonstrate clear improvement in strength and crack resistance of the mixtures with the additives. Optimization based on response surface methodology to determine optimal mixture proportions is a strength. The XRD and SEM analyses provide good insights into the microstructural improvements. Overall, it is a technically sound study contributing good knowledge to promote sustainable use of recycled aggregate and industrial byproducts in pavement base construction.
Comments:
- Abstract: Define alkali-activated lithium slag and fly ash for readers new to the topic.
- Expand on issues with directly discarding waste concrete slabs.
- Provide more details on the source and production of lithium slag.
- Can you explain the micro-expansion phenomenon?
- Consider defining two-dimensional contour maps and three-dimensional response surface plots.
- When mentioning the issues of directly discarding waste concrete slabs, also highlight the associated economic losses if not recycled and reused efficiently in new construction.
- Provide more background details on the production and composition of lithium slag used. Is it a byproduct of lithium extraction from ores? What is its typical chemical composition? This will help readers better understand its role.
- When mentioning the micro-expansion phenomenon due to alkali-activated lithium slag, provide a 2-3 line explanation of the underlying chemical reactions causing this expansion and how it benefits shrinkage reduction.
- Define two-dimensional contour maps and three-dimensional response surface plots in 1-2 lines for readers who may be unfamiliar with these terms. Briefly mention what useful insights they provide.
- In Figure 1 legends, increase the font size for the bar chart categories (40% RCA Control, 10% fly ash etc.) to improve readability.
- In Figure 3, the axis labels, grid lines and contour lines are unclear. Please improve overall image quality and resolution for better reproduction.
- Define the diffraction angle term (2θ) when first introduced here and explain what useful information it provides regarding material phases.
- Rephrase "...determined as 15%, 10%, and 40%, respectively" to improve clarity.
- Provide 2-3 examples of other relevant industrial byproducts that can be explored as additives based on this study.
- Practical implications section: Add a paragraph on feasibility of field implementation and expected economic benefits vs issues to be resolved before adopting these mixes in road projects.
- Test methods section: Provide more specifics on curing conditions used for strength and shrinkage test specimens - temperature, humidity levels and duration.
- Results section, page 4: Correct the typographical error of using "crack" instead of "shrinkage" when referring to shrinkage performance.
Minor editing of English language required
Author Response
We thank the reviewers for providing constructive feedback. We have fully revised our manuscript (minerals-2821851) and have addressed all of the reviewers’ comments, as well as added new analyses to further strengthen our work.
Point-by-point responses please see the attachment.

Reviewer 2 Report
Comments and Suggestions for Authors
In this manuscript, the effects of adding alkali-activated lithium slag, fly ash, and their combination on the compressive strength, splitting strength, and shrinkage performance of recycled cement stabilized macadam mixture were investigated. Therefore, this research aimed to is to improve the working performance of recycled cement stabilized macadam base and provide industrial solid waste utilization. However, before giving further consideration to publication, it is recommended that the following observations be fully addressed.
Comment 1: In the abstract, AFt is ettringite, not calcite.
Comment 2:The major defect of this study is the debate or Argument is not clear stated in the introduction session. Hence, the contribution is weak in this manuscript. I would suggest the author to enhance your theoretical discussion and arrives your debate or argument.
Comment 3: Please underscore the scientific value added of your paper in your abstract and introduction.
Comment 4: Following, you will find some new related references which should be added to literature review:
1. Zhang et al. Properties of RCA stabilized with alkali-activated steel slag based materials in pavement base: Laboratory tests, field application and carbon emissions. https://doi.org/10.1016/j.conbuildmat.2023.134547.
2. Wang et al. The influence of fly ash dosages on the permeability, pore structure and fractal features of face slab concrete. https://doi.org/10.3390/fractalfract6090476.
3. Wang et al. Pore structural and fractal analysis of the influence of fly ash and silica fume on the mechanical property and abrasion resistance of concrete. https://doi.org/10.1142/S0218348 X21 40003X.
Comment 5: How the experimental design of this paper is carried out needs to be explained in detail.
Comment 6: In line 123, “3.Materials and Methods”is wrong. Should be the result and discussion.
Comment 7: In this paper, the response surface method is used to study the optimal content of lithium slag, fly ash and recycled aggregate, but the experimental design scheme of this method and the level of test factors are not provided. In addition, the optimal incorporation of lithium slag, fly ash and recycled aggregate is consistent with the results of single factor control test. What is the significance of doing this research? Please consider.
Comment 8: The data of microscopic analysis and mechanism analysis are difficult to support the conclusion. It is necessary to supplement the sufficient supporting data and add the corresponding reference support.
Comment 9: The SEM image is difficult to fully support the corresponding narrative in the paper.
Comment 10: The overall framework of this paper needs to be re-combed.
Author Response

(The authors gave the same response as above.)

Reviewer 3 Report
Comments and Suggestions for Authors
The paper is original and well-done, congratulations.
Just some suggestions:
1) From line 298, pay attention to subscripts in the text.
2) In consideration of the interesting results, it is recommended to firstly discuss the obtained results and then to go into detail (e.g. with a bulleted list). The conclusions can be improved.
Best Regards
Author Response

(The authors gave the same response as above.)
